# No Training Data, No Cry: Model Editing without training data or Finetuning

## Abstract

Model Editing(ME)–such as classwise unlearning and structured pruning–is a nascent field that deals with identifying editable components that, when modified, significantly change the model's behaviour, typically requiring fine-tuning to regain performance. The challenge of model editing increases when dealing with multi-branch networks(e.g. ResNets) in the data-free regime, where the training data and the loss function are not available. Identifying editable components is more difficult in multi-branch networks due to the coupling of individual components across layers through skip connections. This paper addresses these issues through the following contributions. First, we hypothesize that in a well-trained model, there exists a small set of channels, which we call HiFi channels, whose input contributions strongly correlate with the output feature map of that layer. Finding such subsets can be naturally posed as an expected reconstruction error problem. To solve this, we provide an efficient heuristic called RowSum. Second, to understand how to regain accuracy after editing, we prove, for the first time, an upper bound on the loss function post-editing in terms of the change in the stored BatchNorm(BN) statistics. With this result, we derive BNFix, a simple algorithm to restore accuracy by updating the BN statistics using distributional access to the data distribution. With these insights, we propose retraining free algorithms for structured pruning and classwise unlearning, CoBRA-P and CoBRA-U, that identify HiFi components and retains(structured pruning) or discards(classwise unlearning) them. CoBRA-P achieves at least 50% larger reduction in FLOPS and at least 10% larger reduction in parameters for similar drop in accuracy in the training free regime. In the training regime, for ImageNet, it achieves 60% larger parameter reduction. CoBRA-U achieves, on average, a 94% reduction in forget-class accuracy with a minimal drop in remaining class accuracy.[1]

## 1 Introduction

The improved performance of deep learning models on various tasks (Krizhevsky et al., 2012; Ioffe & Szegedy, 2015; He et al., 2016) has increased their adoption. However, such models may not always be suitable for direct use in various applications. For instance, a pre-trained classification model might not run on an edge device without compressing it using a technique such as pruning (Prakash et al., 2019). We use the term *Model Editing* to refer to such modifications.

This work focuses on two model editing tasks - pruning and classwise unlearning for vision tasks. Pruning (LeCun et al., 1989; Hoefler et al., 2021) is one of the methods to improve latencies and memory requirements of models during inference. Pruning involves discarding "unimportant" components of a model, such as weights, neurons, or channels. This work focuses on structured pruning (Luo et al., 2017; Wang et al., 2020b; Shen et al., 2022) that discards entire channels in Convolution Neural Networks (CNNs) as opposed to unstructured pruning (LeCun et al., 1989; Han et al., 2015; Tanaka et al., 2020) that discards weights individually. Classwise unlearning (Jia et al., 2023) refers to the task where the goal is to unlearn training data points of an entire class while maintaining the predictive performance on remaining classes. Classwise unlearning can be efficiently performed using pruning (Jia et al., 2023).

Editing tasks such as pruning and classwise unlearning require an understanding of the components

---

[1]The code is available at `https://anonymous.4open.science/r/cobra-197B`

of a model - such as weights, neurons, or convolutional filters that contribute significantly to its prediction (Räuker et al., 2023). This becomes more challenging when dealing with modern neural networks that consist of skip connections (He et al., 2016; Huang et al., 2017) that couple elements between layers (Liu et al., 2021a; Fang et al., 2023). However, this is not generally addressed in relevant works (Jia et al., 2023; Ding et al., 2021; Joo et al., 2021; Luo et al., 2017). Editing algorithms often take a toll on the original task performance and thus rely on retraining to alleviate this (Luo et al., 2017; Wang et al., 2020b; Jia et al., 2023).

However, retraining requires significant computational resources and access to the loss function & training set pertaining to the original task. It is not uncommon for the relevant training set & loss function to be unavailable due to privacy or commercial concerns (Yin et al., 2020), making retraining more challenging. Most existing works either assume access to data and finetune models (Jia et al., 2023; Wang et al., 2020b; Shen et al., 2022) or assume the absence of training data and do not finetune (Narshana et al., 2022; Murti et al., 2022; Tanaka et al., 2020). However, the gap between the accuracy of data-free and data-driven methods is significant (Hoefler et al., 2021). Thus, it is important to bridge this gap.

For model editing, this work, similar to Murti et al. (2022), assumes access to samples with similar distributional properties to that of the training set. For instance, to construct a cat-dog classifier, a training set could be a large collection of images of cats and dogs taken from a private image repository, while samples available via distributional access could be the photos of cats and dogs taken from a personal device. We use this distributional information to study CNNs with Batchnorm layers (Ioffe & Szegedy, 2015). Batch Normalization, a popular deep learning technique developed to decrease training time, is used in many successful architectures like ResNets (He et al., 2016), VGGs (Simonyan & Zisserman, 2015), and MobileNet (Howard, 2017). Existing theoretical analysis of Batch Normalization has focused on understanding its effect during training (Santurkar et al., 2018); however, to the best of our knowledge, there has been little insight into its effect on the loss function during inference upon model perturbation. Towards addressing the challenges presented above, the following are our contributions:

1. It is important for model editing to understand what components of a well-trained model are necessary for predictions. To address this, we propose the notion of High-Fidelity(HiFi) components, components of the network that contribute significantly to the output of the corresponding layer. Using this notion, we hypothesize that in each layer of a well-trained model, the set of HiFi components are responsible for the model's performance, which we empirically validate in Section 7. Thus, the problem of model editing boils down to identifying HiFi components.

2. Towards identifying HiFi components in a layer for model editing without access to training data or the loss function, We use correlation as the measure of similarity between the distribution of the input channel's contribution to the output and the distribution of the output. In Section 4, we show that this choice of similarity naturally connects HiFi components to those with low expected reconstruction error, a popular saliency measure in pruning. However, this problem is NP-Hard, and the use of a heuristic called RowSum is required to solve this problem. This enables the identification of editable components using distributional access.

3. Typically, editing causes a degradation in the model's performance. To understand the impact of BatchNorm parameters on this degradation, we derive a connection between the learned parameters of BatchNorm layers and the loss function. We show that the loss function can be upper bounded by a quadratic function of the learned parameters of the BatchNorm layer. We state this formally in Theorem 1. Based on our analysis, we propose Algorithm 2, called BNFix, an algorithm requiring only distributional access to modify the stored statistics in a BatchNorm layer to reduce performance degradation due to model editing. We observe an interesting phenomenon, which we call **BN Recall**, when applying BNFix as a replacement for retraining using remaining class examples - applying BNFix on a model whose forget accuracy has significantly fallen using only remain class samples causes the forget class accuracy to increase significantly.

4. In addition to identifying HiFi components and BNFix, we use fidelity compensation - where we improve the fidelity of the feature maps via weight rescaling - to design the CoBRA family of editing algorithms and analyze this improvement in Theorem 2. CoBRA(Correlation-based editing with Batchnorm Re-Adjustment) is an editing scheme that identifies HiFi components in each layer of a network to either retain(CoBRA-P) or discard(CoBRA-U), and recovers model performance by BNFix and weight compensation. Our experiments show that CoBRA-P achieves at least 50% larger reduction in FLOPS and at least 10% larger reduction in parameters for similar

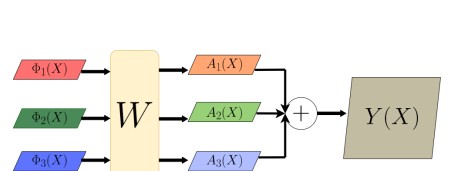
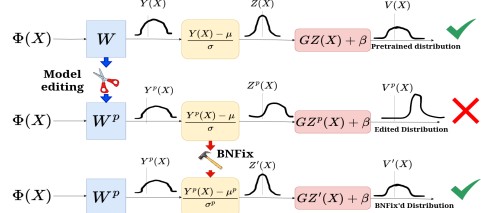

**(a)** How input contributions generate feature maps:

**(b)** Editing models and the role of BatchNorm

**Figure 1: Left Image:** Each channel of the input features generates an *input contribution*, which are then summed to obtain the feature map. **Right Image:** After editing a layer, feature distributions of subsequent layers are changed; adjusting BN stats helps address this.

drop in accuracy in the training free regime. In the training regime, for ImageNet, it achieves 60% larger parameter reduction. CoBRA-U achieves, on average, a 94% reduction in forget-class accuracy with a minimal drop in remain class accuracy.

## 2 PRELIMINARIES

**Notation** Let $\boldsymbol{a} \in \mathbb{R}^n$ denote an $n$-dimensional vector whose $i^{th}$ element is $a_i$, and $\boldsymbol{B} \in \mathbb{R}^{n \times m}$ a matrix with $n$ rows and $m$ columns whose $i^{th}$ row is $\boldsymbol{b}_i \in \mathbb{R}^m$. For $p \in \mathbb{N}$, let $[p] = \{1, \ldots, p\}$. For matrices, $\boldsymbol{A}, \boldsymbol{B} \in \mathbb{R}^{n \times m}$, we define $\langle \boldsymbol{A}, \boldsymbol{B} \rangle = \text{Tr}(\boldsymbol{A}^\top \boldsymbol{B})$ and frobenius $\|\boldsymbol{A}\|_F^2 = \langle \boldsymbol{A}, \boldsymbol{A} \rangle$. For tensors $\mathbf{A}, \mathbf{B} \in \mathbb{R}^{C \times K \times K}$, we define $\langle \mathbf{A}, \mathbf{B} \rangle = \sum_{i=1}^C \langle \boldsymbol{A}_i, \boldsymbol{B}_i \rangle$, where $\boldsymbol{A}_i, \boldsymbol{B}_i \in \mathbb{R}^{K \times K}$ and $\|\mathbf{A}\|^2 = \langle \mathbf{A}, \mathbf{A} \rangle$. For a vector $\boldsymbol{v}$, $\text{diag}(\boldsymbol{v})$ is a diagonal matrix whose $i^{th}$ entry is $v_i$. $\text{Top}_p(\boldsymbol{v})$ denotes a function that which returns the indices of the elements in the top $p^{th}$-percentile of $\boldsymbol{v}$.

**Neural Network Preliminaries** Let $f_\theta$ be a neural network with parameters $\theta$ with $L$ layers. Consider data drawn from a distribution $\mathbb{P}_\mathcal{D}$, we use X as a random variable drawn from this distribution. We use $\mathcal{L}_\theta(\boldsymbol{x})$ as the loss function evaluated with parameters $\theta$ on a point $\boldsymbol{x}$ and parameters are trained to minimize the expected loss over the distribution. The parameters are grouped into structural units, such as convolutional filters in CNNs, and are stacked in layers. We refer to such structures as *components* of the network. The structures and the operations performed on the input by these structures form the architecture of the network.

**2D Convolution** Let the $l^{th}$ layer of a network be a 2D convolution layer with $c_{in}^l$ input channels and $c_{out}^l$ output channels whose weights are $\mathbf{W}^l \in \mathbb{R}^{c_{out}^l \times c_{in}^l \times k \times k}$, where $k$ is the kernel size. Let the input to the convolution layer be $\Phi^l(\boldsymbol{x}) \in \mathbb{R}^{c_{in}^l \times h^{l-1} \times w^{l-1}}$, and the output $\mathbf{Y}^l(\boldsymbol{x}) \in \mathbb{R}^{c_{out}^l \times h^l \times w^l}$, where $h^{l-1}, h^l$ and $w^{l-1}, w^l$ represent the heights and widths of the input and output respectively. The $c^{th}$ output channel, $\boldsymbol{Y}_c^l$ is then,

$$\boldsymbol{Y}_c^l(\boldsymbol{x}) = \sum_{i=1}^{c_{in}^l} \boldsymbol{\Phi}_i^l(\boldsymbol{x}) * \boldsymbol{W}_{ci}^l = \sum_{i=1}^{c_{in}^l} \boldsymbol{A}_{ci}^l(\boldsymbol{x}) \tag{1}$$

where $*$ denotes the convolution operation. We say $\boldsymbol{A}_{ci}^l(\boldsymbol{x}) \in \mathbb{R}^{h^l \times w^l}$ is the *input contribution* of input channel $i$ to output channel $c$; this is illustrated in Figure 1a.

**Batch Normalization during inference** Let the $l^{th}$ layer of a neural network be a a BatchNorm layer with dimension $m$ whose input is $\boldsymbol{y}^l(x) \in \mathbb{R}^m$, parameterized by two stored statistics, mean $\boldsymbol{\mu} \in \mathbb{R}^m$ and standard deviation $\boldsymbol{\sigma} \in \mathbb{R}^m$, and two learned parameters, shift $\boldsymbol{\beta} \in \mathbb{R}^m$ and scale $\boldsymbol{\gamma} \in \mathbb{R}^m$. The $c^{th}$ output of the layer during inference, $\boldsymbol{v}^l(\boldsymbol{x}) \in \mathbb{R}^m$, is given by

$$\boldsymbol{v}^l(\boldsymbol{x}) = \boldsymbol{G}\boldsymbol{z}^l(\boldsymbol{x}) + \boldsymbol{\beta} \quad \text{where} \quad z_c^l(\boldsymbol{x}) = \frac{y_c^l(\boldsymbol{x}) - \mu_c}{\sigma_c} \tag{2}$$

where $\boldsymbol{G} = \text{diag}(\boldsymbol{\gamma})$. The stored statistics are meant to estimate the mean and standard deviation of $\boldsymbol{y}^l(\text{X})$ from the *training data*. Additional details are in Appendix C.

## 3 THE PROBLEM OF EDITING WELL-TRAINED MODELS WITHOUT TRAINING DATA

Model editing refers to techniques that selectively change the model parameters to modify its statistical behaviour (Jia et al., 2023; Santurkar et al., 2021; Shah et al., 2024), motivated by issues such as privacy and GDPR regulations (Bourtoule et al., 2021; Nguyen et al., 2022). Editing encompasses a wide variety of tasks, including debiasing (Jain et al., 2022), selective unlearning (Golatkar et al., 2020), network scrubbing (Kurmanji et al., 2024), and lifelong learning (Sahoo et al., 2024; Golkar et al., 2019). Recently, *component attribution* - that is, identifying components responsible for predictions - has gained traction for model unlearning (Shah et al., 2024; Wang et al., 2022; Kodge et al.). However, it is challenging to use model editing without the loss function and training data (Shah et al., 2024), as well as for analyzing models with complex interconnections (Narshana et al., 2022; Liu et al., 2021a). Extensive related work is cited in Appendix A. In this section we formalize the problem of Model Editing via pruning.

**What is Model Editing?** Consider the model $f_{\theta_0}$, and let $\mathcal{D}_i$, $i \in [M]$ be conditional data distributions, such as classes. Our goal is to *edit* the model by removing entire components. That is, given the weights of the well-trained model $\theta_0$, we edit $\theta_0$ to $\theta_E = \theta_0 - \theta^\star$, where $\theta^\star \in S_B := \{\theta \in \mathbb{R}^d : \text{count}(f_{\theta_0 - \theta}) = B\} \subset \mathbb{R}^d$ by *editing the parameters of at most $C_{total} - B$ components, where $C_{total}$ is the total number of components in the network (i.e., convolutional filters) by solving*

$$\theta^\star = \underset{\theta \in S_B}{\arg\min} \sum_i \mathbb{E}_{X \sim \mathcal{D}_i} \left[ \alpha_i \left( \mathcal{L}_{\theta_0 - \theta}(X) - \mathcal{L}_\theta(X) \right) \right], \tag{Edit}$$

where $\alpha_i \in \mathbb{R}$ are multipliers to weight tasks, depending on whether we want the model to increase the loss or decrease it on the corresponding distribution $\mathcal{D}_i$. While a variety of tasks can be classified as model editing (Shah et al., 2024); in this work, we address the problems of **structured pruning** and **classwise unlearning.**

**Structured Pruning**, in the setting of equation Edit, is when $M = 1$, and $\alpha_1 = 1$. Thus, we write

$$\theta^\star = \underset{\theta \in S_B}{\arg\min} \ \mathbb{E}_{X \sim \mathcal{D}} \left[ \left( \mathcal{L}_{\theta_0 - \theta}(X) - \mathcal{L}_\theta(X) \right) \right], \tag{Prune}$$

**Classwise unlearning** involves removing the model's ability to make accurate predictions on a chosen class, called the **forget class** with distribution $\mathcal{D}_f$, while maintaining the statistical performance on the remaining classes - called the **remain classes**, with distribution $\mathcal{D}_r$. In the setting of equation Edit, we have $M = 2$, $\mathcal{D}_1 = \mathcal{D}_f$, $\mathcal{D}_2 = \mathcal{D}_r$, $\alpha_1 = -1$ and $\alpha_2 = \kappa > 0$. Solving this problem ensures that the loss on $\mathcal{D}_f$ increases, while the loss on $\mathcal{D}_r$ decreases, with $\kappa$ penalizing the extent to which $\mathbb{E}_{X \sim \mathcal{D}_f} \left[ \mathcal{L}_{\theta_0 - \theta}(X) \right]$ is allowed to increase. We write this as

$$\theta^\star = \underset{\theta \in S_B}{\arg\min} \, \mathbb{E}_{X \sim \mathcal{D}_r} \left[ \kappa \left( \mathcal{L}_{\theta_0 - \theta}(X) - \mathcal{L}_\theta(X) \right) \right] - \mathbb{E}_{X \sim \mathcal{D}_f} \left[ \mathcal{L}_{\theta_0 - \theta}(X) - \mathcal{L}_\theta(X) \right]. \tag{Forget}$$

**Challenges in editing models without the training data or loss function?** Unlike works such as Jia et al. (2023) and the references therein, fine-tuning or retraining the model is not possible in this setting. Thus to effectively edit the behavior of a network, it is necessary to identify the components that are responsible for making predictions. These can be characterized as components which when modified, significantly change the behaviour of the network. The key challenge is thus:
**Problem Statement: Solve equation Prune or equation Forget without access to original training data and loss function which was used to obtain $\theta_0$.**

It is well known that pruning or perturbing a large number of components significantly affects statistical performance (Hoefler et al., 2021). Thus, it is necessary to identify a *small subset of editable components*; components which are **editable** can be removed to aid an editing task. In the case of pruning, components that have no effect on the performance of the model are editable, whereas for model unlearning, components required only for the prediction of the forget class are editable. We use this insight to pursue the stated problem and develop algorithms to address it.

## 4 INDENTIFYING EDITABLE COMPONENTS THROUGH HIFI COMPONENTS

As stated in the previous section, editing well-trained models without access to the training data or loss function requires identifying components that have a disproportionate impact on the models's

predictive performance. In this section, we propose the notion of High-Fidelity (HiFi) components, and hypothesize that HiFi components are what govern a model's predictive performance. We empirically validate our hypothesis and provide a template for model editing algorithms derived from it.

## 4.1 WHICH FEATURES ARE DISTRIBUTIONALLY SIMILAR TO THE OUTPUT FEATURES?

We provide the empirical observation that in many layers of deep networks, there are only a few filters for which the input contribution distribution is similar to that of the output distribution. In Figure 2, we show the relative reconstruction error after removing filters from a selection of layers of a ResNet50 trained on CIFAR10 - we use the expected reconstruction error as a measure of distributional similarity. We see that in well-trained models, a small subset of filters - between 5% and up to 30% of the number of filters in the layer - generate input contributions that are distributionally similar to the aggregate feature maps. This observation motivates us to edit models *by identifying those components whose input contributions are distributionally similar to the feature maps.* We call such components **High Fidelity (HiFi) components**, which we define in the sequel.

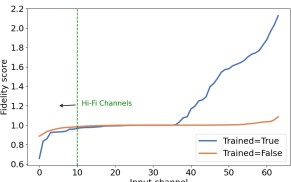

(a) Reconstruction error for layer 1, block 0, conv 2.

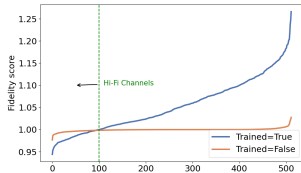

(b) Reconstruction error for layer 4, block 2, conv 2

**Figure 2:** Comparision of the fidelity scores of two different layers

## 4.2 HIGH FIDELITY COMPONENTS AND THE FIDELITY SCORE

Suppose $Y^{l+1}(X)$ is the feature map generated by layer $l+1$, and suppose $A_i^{l+1}(X)$ is the $i$th input contribution, as defined in equation 1. We say the $i$-th component in layer $l$ is a **high-fidelity (HiFi) component** if the distribution of the input contribution $A_i^{l+1}(X)$, $\mathcal{D}_i^{l+1}$ in layer $l+1$ is similar to the distribution of $Y^{l+1}(X)$, $\mathcal{D}^{l+1}$. HiFi components are those with input contributions that can reconstruct the aggregate feature map[2]. To capture this, we analyze the dissimilarity between the distributions of $\hat{Y}^{l+1}(X) = Y^{l+1}(X) - \mathbb{E}_X\left[Y^{l+1}(X)\right]$ and $\hat{A}_i^{l+1}(X) = A_i^{l+1}(X) - \mathbb{E}_X\left[A_i^{l+1}(X)\right]$. We define $\mathsf{FS}(i)$, a *Fidelity score* that measures the similarity between an input contribution and the aggregate feature map, below.

$$\mathsf{FS}(i) = \mathsf{DIS}(\hat{\mathcal{D}}^{l+1}, \hat{\mathcal{D}}_i^{l+1}) = \left( \frac{\mathbb{E}_X\left[\|\hat{Y}^{l+1}(X) - \beta_i \hat{A}_i^{l+1}(X)\|^2\right]}{\mathbb{E}_X\left[\|\hat{Y}^{l+1}(X)\|^2\right]} \right)^{\frac{1}{2}} \tag{3}$$

where $\beta_i = \mathbb{E}_X\left[\langle \hat{Y}^{l+1}(X), \hat{A}_i^{l+1}(X)\rangle\right]\mathbb{E}_X\left[\|\hat{A}_i^{l+1}(X)\|\right]^{-2}$

In the above definition, the smaller the value of $\mathsf{DIS}(\hat{\mathcal{D}}^{l+1}, \hat{\mathcal{D}}_i^{l+1})$ (or higher the value of $\beta_i$) is, better the reconstructability of $Y$ in the mean-square sense. Furthermore, note that we can apply equation 3 on a channel-by-channel basis by considering the distributions of a single output feature map in a layer; we add an the additional subscript $c$ to indicate that the feature map (and the input contribution) are generated by the $c$th component in the layer. In well-trained models we often observe that a small number of components have relatively lower $\mathsf{DIS}$ scores than the rest. Identifying such components is key to understanding the statistical behavior of model outputs, and hence will be the most critical insight for the subsequent development of our algorithms.

---

[2]This motivates the name HiFi components: Components whose sum can accurately reconstruct the output with Hi-Fidelity

**The Role of $\beta_i$ and RowSum:** $\beta_i$ is a variant of the Tensor correlation between the input contribution $A_i^{l+1}$ and the feature map $Y^{l+1}$. Furthermore, we can show that

$$\mathsf{DIS}(\hat{\mathcal{D}}^{l+1}, \hat{\mathcal{D}}_i^{l+1})^2 = \mathbb{E}_X\left[\|\hat{Y}^{l+1}(X)\|^2 - \beta_i^2\|\hat{A}_i^{l+1}(X)\|^2\right]/\mathbb{E}_X\left[\|\hat{Y}^{l+1}(X)\|^2\right] \qquad (4)$$

highlights the relation between $\mathsf{FS}(i)$ and $\beta_i$ - if $\|\hat{A}_i^{l+1}(X)\|$ is roughly equivalent for all $i$, then $\mathsf{FS}(i)$ is low when $\beta_i$ is large. Thus, a heuristic for identifying HiFi components is finding components for which $\beta_i$ is large. Moreoever, note that $\beta_i$ can be written as the sum of the elements of the row of a matrix, motivating the naming of the heuristic **RowSum**. Specifically, $\beta_i\mathbb{E}_X\left[\|\hat{A}_i^{l+1}(X)\|^2\right] = \sum_j Q_{ij}$, where $Q_{ij} = \mathbb{E}_X\left[\langle \hat{A}_i^{l+1}, \hat{A}_j^{l+1}\rangle\right]$. We examine this in greater detail in Appendix E, along with an examination of the reconstruction error after the BatchNorm layers. Based on our empirical observations that a small subset of components in well-trained models generate input contributions that are distributionally similar to the feature maps, we now state the main hypothesis of our work. We validate our hypothesis empirically in Section 7.

**Hypothesis 1.** *Suppose we have a well-trained model with parameters $\mathcal{W} = (W_1, \cdots, W_L)$. We hypothesize that the HiFi components of this model contribute most to the predictions of the model, and those components that are not high fidelity can be discarded without affecting the performance of the model.*

**Using HiFi Components for Model Editing** Hypothesis 1 states that only the HiFi components - a small subset of the components in a layer - are responsible for the model's predictions. Thus, it facilitates model editing as the distributional similarity between input contributions and aggregate feature maps, as measured using equation 3, can be used as a surrogate for the impact of removing that component on the loss function. Thus, leveraging this hypothesis, we can either *prune* the HiFi components to increase the loss (for instance, for classwise unlearning tasks), or *retain* them to ensure the loss remains low (for instance, for structured pruning). We provide a generic algorithmic recipe for model editing using HiFi components, specialized for the tasks of classwise unlearning and structured pruning in Algorithm 1; these are discussed in greater detail in Section 6.

---

**Algorithm 1:** Model Editing by Identifying HiFi Channels

**Input:** Model $f_{\theta_0}$ with layer indices $[L]$, layerwise budgets $\{B_l\}_{l\in[L]}$, data distributions $\mathcal{D}_1$, $\mathcal{D}_f, \mathcal{D}_r$
**Output:** Edited model $f_{\theta_E}$
**for** $l \in [L]$ **do**
    **Compute** $\mathsf{FS}(i)$ using equation 3 on $\mathcal{D}, \mathcal{D}_f$, $\mathcal{D}_r$
    **Determine** which components to edit
**Recover** accuracy on $\mathcal{D}$

---

## 5 BNFIX: AN ALTERNATIVE TO RETRAINING BY RESETTING BN STATISTICS

In this section, we analyze BatchNorm1D in single branch networks during inference and how the change in distribution due to editing affects the relationship between the loss and BatchNorm parameters. Using this, we derive an algorithm to correct stored statistics after editing. This update has been previously employed in pruning literature (Frantar et al., 2022), but to the best of our knowledge, this is the first work to provide theoretical basis to the update in a distributional setting.

**Analysis of BatchNorm at Inference** BatchNorm at inference shifts the distribution of the intermediate representation at the output of a layer to have mean $\boldsymbol{\beta}$ and standard deviation $\boldsymbol{\gamma}$. These are parameters of the model which are minimize a loss function $\mathcal{L}$ as described in Section 2 .We use the following fact to analyze the loss in terms of the intermediate representation at the output of a layer.

**Fact 1 (Stochastic Mean Value theorem).** *Let $f$ be a twice differentiable real valued function from $\mathbb{R}^d$ to $\mathbb{R}$ and $\boldsymbol{H}_f(\boldsymbol{x})$ be the Hessian at any $\boldsymbol{x} \in \mathbb{R}^d$. For any point $\boldsymbol{c} \in \mathbb{R}^d$ and a multivariate random variable $\mathrm{X} \in \mathbb{R}^d$ with finite second order moments, there exists a random variate $\mathrm{t} \in (0,1)$ such that*

$$f(\mathrm{X} + \boldsymbol{c}) = f(\boldsymbol{c}) + \nabla f(\boldsymbol{c})^\top \mathrm{X} + \frac{1}{2}\mathrm{X}^\top \boldsymbol{H}_f(\boldsymbol{c} + \mathrm{t}\mathrm{X})\mathrm{X}$$

For a proof, see Corollary 2 in Yang & Zhou (2021). Though for the case discussed here the above fact suffices, but one could potentially use similar facts which can be deduced from other techniques such as Delta Method (Benichou & Gail, 1989) or obtain a result on the expectation such as (Massey

& Whitt, 1993). Using the optimality of the learned parameters of BatchNorm and Fact 1, we make some assumptions on the first and second-order derivatives on the loss for a well trained model in terms of the learned parameters of BatchNorm layers.

**A.1** For a well-trained model $\nabla\mathcal{L}(\beta) = 0$, the gradient with respect to the shift parameter is zero.

**A.2** For a fixed $G$ and any $\beta$, we can bound the eigenvalues of the hessian with a constant $K$ for all inputs. Formally, $\|H_\mathcal{L}(GZ + \beta)\|_2 \leq K$ for all random variables $Z \in \mathbb{R}^d$ such that $E(Z_i^2) = 1, E(Z_i) = 0$ for all $\beta \in \mathbb{R}^d$. Here, the norm is the spectral norm of a matrix.

The assumptions **A.1** and **A.2** capture the model's "well-trained-ness" on the objective function $\mathcal{L}$ and follow from the first and second-order necessary conditions of optimality. We note that **A.1** would not hold if the input distribution to the network was different from that of the training distribution. The constant $K$ captures the smoothness of the loss function with respect to the parameter $\beta$ and subsumes the effect of the rest of the network, which may contain more linear and non linear layers. Equipped with these assumptions about well-trained models, we derive a bound on the average loss over the learned distribution in terms of the learned parameters of the BatchNorm layer. With the observation that $\mathbb{E}[V(X)] = \beta$, the term $\mathcal{L}\left(\mathbb{E}[V(X)]\right) = \mathcal{L}(\beta)$ reflects the loss of the averaged intermediate representation.

**Lemma 1** (Loss of a well trained model expressed with BatchNorm). *Consider a model that satisfies assumptions 5. We can express an upper bound on the expected loss during inference in terms of the statistics of the output of the BatchNorm layer $V(X) \in \mathbb{R}^m$.*

$$|\mathbb{E}[\mathcal{L}(V(X))] - \mathcal{L}(\beta)| \leq \frac{K}{2}||\gamma||^2 \tag{5}$$

*proof sketch.* We prove this with fact 1 and using the statistics of $V(X)$. A full proof of Lemma 1 can be found in D.1. $\square$

**How Editing affects BatchNorm** We now study how editing affects the statistics of the output of the batch norm layer and the loss. Using lemma 1, we analyse the effect on the objective $\mathcal{L}$ due to the change in the intermediate distribution to state Theorem 1. It shows that the loss is upper bounded by a quadratic function of the difference of the mean of the distribution and ratio of the variances. This allows us to qualitatively measure the effect of the shift in distribution on the loss function.

---

**Algorithm 2:** BNFix

**Input:** Batch Norm Layer $l$ with $m$ channels, dataset $\mathcal{D} = \{X_i\}_{i=1}^N$

**for** $c \in [m]$ **do**
$\quad \mu_c^l \leftarrow \frac{1}{N}\sum_{b=1}^N Y_c^l(X_b);$
$\quad \sigma_c^{2\,(l)} \leftarrow \sum_{b=1}^N \frac{(Y_c^l(X_b)-\mu_c^l)^2}{N-1};$

---

**Theorem 1.** *Let the $l^{th}$ layer of a network be a BatchNorm layer as described in 2 with stored data statistics $\mu_c$ and $\sigma_c^2$. Editing components of preceeding layers causes a change in the distribution of the intermediate representation to some $Y^{(p)}(X)$, with modified moments $\mu^{(p)}$ and $(\sigma^{(p)})^2$. The output of BatchNorm after editing is then, $\mathbf{V}^{(p)} = GZ^{(p)} + \beta$ where $Z^{(p)} = \frac{Y_c^{(p)}(X)-\mu_c}{\sigma_c}$. Then,*

$$|\mathbb{E}[\mathcal{L}(\mathbf{V}^{(p)}(X))] - \mathcal{L}(\beta)| \leq \frac{K}{2}\left(\sum_{i=1}^d \gamma_i^2\left(\left(\frac{\sigma_i^{(p)}}{\sigma_i}\right)^2 + \left(\frac{\mu_i^{(p)} - \mu_i}{\sigma_i}\right)^2\right)\right) \tag{6}$$

*proof sketch.* We prove this result using the properties of normalization and apply Lemma 1. The full proof of this theorem can be found in D.1. $\square$

Based on Theorem 1, we observe that updating stored statistics to represent the new moments of the intermediate representations after editing, i.e., setting $\mu_i = \mu_i^{(p)}$ and $\sigma_i = \sigma_i^{(p)}$, restores the upper bound on the loss function to Lemma 1. However, the bound suggests that only channels for which the coefficient of $\gamma_i^2$ in equation 6 is greater than 1 should be updated to decrease the upper bound. We study this in Appendix B.9 and emperically show that updating the statistics of all channels leads to larger accuracy recovery in the case of pruning. Algorithm 2 shows the update procedure for the stored statistics of a single batch norm layer. This gradient-free procedure does not require training samples and can be implimented using a small number of samples obtained from distributional access. In Appendix B.2, we display the effectiveness of the algorithm on a simple synthetic task.

# 6 MODEL EDITING THROUGH CORRELATION STRUCTURE OF COMPLEX INTERCONNECTIONS

A key challenge in applying the HiFi hypothesis 4 is identify HiFi components across groups of interconnected layers in complex networks. We propose Algorithm 3 to identify HiFi components over all layers in a DFC to extend the HiFi hypothesis to networks with complex interconnections.

**Computational cost of Algorithm 3.** Let $N$ be the number of data points used to estimate the saliency and $M^l$ be the complexity of computing the input contribution at layer $l$ for a single sample in a DFC with $m$ layers. The complexity to compute the set of HiFi channels for an output channel of a layer is, $t_{sal}^l = O(NM^l C_{in}^l d^l)$. To select the HiFi components for the DFC, the top $p$ elements for each layer

---

**Algorithm 3:** Compute HiFi channels over Coupled channels

---

**Input:** Model, keepRatio $p$, Samples $\mathcal{D} = \{X_i\}_{i=1}^N$
**Output:** HiFi channels
**Function** ComputeHiFiSet (*Coupled Channels CC*, $p$, $\mathcal{D}$):
    **for** layer $l \in CC$ **do**
        **for** $o \in C_{out}^l$ **do**
            Compute $R_o^l$ according to Equation
            **RowSum**;
            $H_o^l \leftarrow \text{Top}_p(R_o^l)$;
    **return** $\bigcup_{l,o \in [C_{out}^l]} H_o^l$;

---

and output channel in the DFC are collected, this costs $O(\sum_{l=1}^m C_{out}^l(C_{in}^l \log C_{in}^l + t_{sal}^l))$. We compare this with the BGSC algorithm (Narshana et al., 2022) which has a quadratic dependence on the number of layers in the network, as opposed to the proposed work which is linear in the number of layers.

**Fidelity Compensation by Weight Rescaling** In order to improve the model's performance *without* fine-tuning, we propose a distributional approach to modifying the weights to regain accuracy, by modifying the weights of layer $l+1$ after pruning layer $l$ (similarly, we can modify the weights of *feed out layers* after pruning the feed-in layers of a DFC). Unlike prior work which modifies the weights of entire filters with a single parameter (Xie et al., 2021; Halabi et al., 2022), our result modifies the weights of individual convolutional kernels, thereby granting a more fine-grained approach to weight compensation. First, we define the reconstruction error as follows.

$$\text{RE}_c^{l+1}(v) = \mathbb{E}[\|Y_c^{l+1}(X) - \sum_{i \in [C_{in}]} v_i \Phi_i^l(X) W_{ci}^{l+1}\|^2] \tag{7}$$

where $v \in \mathbb{R}^{C_{in}}$. With this definition, we state the solution to the post-pruning fidelity compensation problem, and the reconstruction error improvement in Theorem 2.

**Theorem 2.** *Let $s^l \in \{0,1\}^{C_{in}} = [\mathbf{1}_K; \mathbf{0}_{C_{in}-K}]$, where $\mathbf{1}_K$ is a vector of $K$ ones, and $\mathbf{0}_{C_{in}-K}$ is a vector of $C_{in} - K$ zeros; we ignore the subscripts for brevity in the sequel. Define $\delta_c \in \mathbb{R}^{C_{in}}$ such that $\delta_{ci} = 0$ when $s_i = 0$. We solve $\hat{W}_{ci}^{l+1} = \hat{\delta}_{ci}^{l+1} W_{ci}^{l+1}$, where $\overline{\delta}_{ci}^{l+1} = [\hat{\delta}_{ci}^{l+1}; \mathbf{0}_{C_{in}-K}]$ that satisfies*

$$\hat{\delta}_c^{l+1} = \underset{\delta_c \in \mathbb{R}^K}{\arg\min} \, \text{RE}_c^l([\delta,0]) = P_c^{-1} p_c \quad \text{and} \quad \frac{\text{RE}_c^l(s^l) - \text{RE}_c^l(\overline{\delta}_{C_{in}}^{l+1})}{\text{RE}_c^l(s^l)} \leq 1 - \frac{\|1 - \overline{\delta}_{ci}^{l+1}\|^2}{\kappa(Q_c^{l+1})(C_{in} - K)} \tag{8}$$

*where $\delta_c^\star$ is a vector containing the optimal values of $\delta_{ci}$, $Q_{c,ij} = \mathbb{E}\left[(W_{cj}^{l+1})^\top \Phi_j(X)^\top \Phi_i(X) W_{ci}^{l+1}\right]$, $P_{c,ij} = Q_{c,ij}$ and $p_{c,i} = \mathbb{E}\left[(Y_c^{l+1})^\top \Phi_i^l(X) W_{ci}^{l+1}\right]$ when $s_i, s_j = 1$, and $\kappa(Q_{c,ij})$ denotes the condition number of $Q_{c,ij}$.*

Based on the RowSum heuristic, fidelity compensation scheme 6, and BNFix 5, following the recipe of 1, we develop, **CoBRA**(Correlation based editing with Batchnorm Re-Adjustment), a model editing framework for pruning and classwise forgetting. We provide the key components of our proposed pruning and unlearning algorithm. Detailed algorithms are presented in Appendix B.11.

**CoBRA-P. Compute:** Compute HiFi channels using Algorithm 3 using distributional samples. **Determine:** Retain HiFi components **Recover:** Compute weight compensation according to equation 8 and perform BNFix using distributional samples.

**CoBRA-U. Compute:** Compute HiFi channels using Algorithm 3 using distributional samples from the *forget class*. **Determine:** Discard HiFi components **Recover:** Compute weight compensation according to equation 8 and perform BNFix using distributional samples of the *remain* class.

## 7 EXPERIMENTS

In this section, we present experimental validation of our method on pruning and class unlearning tasks for CNNs with complex interconnections like ResNets to answer the following questions.

(Q1) **HiFi Hypothesis.** Is it true that there is a small set of High-fidelity channels in a well-trained network?
(Q2) **Effectiveness of CoBRA-P.** Does CoBRA-P result in better accuracy-sparsity tradeoff compared to other data-free algorithms?
(Q3) **BNFix replace retraining.** How does BNFix fare against fine-tuning using synthetic samples when pruning models?
(Q4) **CoBRA-U for unlearning.** Is classwise unlearning, as posed by Jia et al. (2023), feasible without fine-tuning? If yes, how does CoBRA-U fare against their method?
(Q5) **Total Recall of BN.** What role do batch norm statistics play in class forgetting, and how can BNFix help in recovering accuracy?

**Datasets and architectures.** We perform experiments on models including ResNet50/101 and VGG19 trained on CIFAR10/100 and ImageNet datasets.
**Distributional Access.** As a proxy for distributional access to data in CIFAR10/100 experiments, we use samples that are synthetically generated using image generation models. Details of synthetically generated samples are available in Appendix B.1. For ImageNet experiments, we use the test split, which contains 100,000 images without labels to identify HiFi channels. Note that test split, as suggested by the name, is not used to evaluate the performance of ImageNet models. For pruning experiments on ImageNet, we perform full retraining instead of BNFix.
**CoBRA Hyperparameters.** We discuss the hyperparameters used for CoBRA-P/U in Appendix B.10
**Validating HiFi hypothesis.** To answer (Q1), we compute the reconstruction error described in equation 3 for 3 different untrained and trained models on CIFAR10 using 1000 samples from the CIFAR10 validation set. We present these sorted values averaged across different trained and untrained models for every layer in Appendix B.12. We make several observations based on these results. First, for most layers, there is a diversity of scores in trained models compared to untrained models, where the scores of all the channels in untrained models are concentrated around a single value. Second, in trained models, there is a small subset of channels, typically less than 10%, which have fidelity scores less than 1. Thus, this validates the HiFi hypothesis, answering (Q1)

### 7.1 PRUNING EXPERIMENTS: EXPLORING (Q2) AND (Q3)

**Baselines.** To compare the performance of CoBRA-P against other data-free methods, we compare against **DFPC** (Narshana et al., 2022), a state-of-the-art data-free structured pruning algorithm for networks with complex interconnections. To gauge the efficacy of BNFix against retraining with distributional access, we compare against $L_2$-norm-based structured pruning, which computes grouped saliencies for a coupled channel based on the $L_2$ norm of the weights of its filters. We *train* the model obtained with $L_2$ norm-based structured pruning *using the synthetic set* for comparison. To the best of our knowledge, these are the only baselines addressing structured pruning of coupled channels in the data free regime.
**Training details.** Details of pre-trained networks and post-training are given in Appendix B.4.
**Results of Pruning Experiments.** Table 1 presents the results of pruning experiments on ResNet-50. We observe that for a similar drop in accuracy in the training-free regime, we gain **at least** 50% larger reduction in FLOPS and at least 10% larger reduction in parameters. In the training regime, we observe that for similar drop in accuracy, CoBRA-P obtains 60% fewer parameters. To answer (Q2), we find that CoBRA-P, for most cases, results in better accuracy-vs-sparsity tradeoff when compared to other data-free algorithms. To answer (Q3), BNFix is able to outperform fine-tuning in some cases using synthetic samples. While in a few cases, it does not, it still leads to a reasonably good performance when compared to no-finetuning.

### 7.2 FORGETTING EXPERIMENTS: EXPLORING (Q4) AND (Q5)

**Metrics.** We report the forget and retain accuracy averaged across 10 classes of the CIFAR10 dataset.
**Additional details.** Experiments with VGG-19 architecture are present in Appendix B.7 where we

| Dataset | Algorithm | Acc.(%) | RF | RP |
|---------|-----------|---------|-----|-----|
| CIFAR10 | Unpruned | 94.99 | 1x | 1x |
| | DFPC (Narshana et al., 2022) | 90.25 | 1.46x | 2.07x |
| | $L_2$ | 15.91 | 4.07x | 4.71x |
| | $L_2$ w/ ST | 90.12 | 4.07x | 4.71x |
| | CoBRA-P(n) | 92.64 | 1.74x | 1.64x |
| | CoBRA-P | **91.02** | **4.07x** | **5.36x** |
| CIFAR100 | Unpruned | 78.85 | 1x | 1x |
| | DFPC | 70.31 | 1.27x | 1.22x |
| | $L_2$ | 16.77 | 1.93x | 1.40x |
| | $L_2$ w/ ST | **73.83** | 1.93x | **1.40x** |
| | CoBRA-P(n) | 72.96 | 1.40x | 1.10x |
| | CoBRA-P | 70.93 | **1.93x** | 1.38x |
| ImageNet | Unpruned | 76.1 | 1x | 1x |
| | ThiNet (Luo et al., 2017) | 71.6 | 3.46x | 2.95x |
| | GReg-2 (Wang et al., 2021) | 73.9 | 3.02x | 2.31x |
| | OTO (Chen et al., 2021) | 74.7 | 2.86x | 2.81x |
| | DFPC | **73.8** | 3.46x | 2.65x |
| | CoBRA-P | 73.25 | **3.60x** | **4.46x** |

**Table 1:** Experiments of CoBRA-P on CIFAR10, CIFAR100 and ImageNet compared with baselines for ResNet-50. ST=Synthetic Training, training using synthetic samples. CoBRA(n) is the CoBRA algorithm without using BNFix or Weight compensation. RF=relative FLOP reduction, RP=relative parameter reduction

| Algorithm | FA(%) | RA. | PR |
|-----------|-------|-----|-----|
| - | 94.99 | 94.99 | - |
| Jia et al. (2023) | 5.54 | **99.11** | - |
| CoBRA-U(0.003)(no BNFix) | **4.22** | 91.131 | 1.0M |
| CoBRA-U(0.003) | 90.61 | 90.629 | 1.0M |
| CoBRA-U(0.2) | 20.90 | 78.786 | 3.63M |

**Table 2:** Class forgetting on CIFAR10 with ResNet-50. CoBRA-U($p$) indicates the hyperparameter for Algorithm 3. For $p = 0.003$, we only prune the last 12 convolution layers and for the last 30 convolution layers for $p = 0.2$. FA=Forget Accuracy, RA = Remain Accuracy, PR=Parameters removed

make similar observations.

**Results of Class-Unlearning Experiments.** We report the results of our algorithm in Table 2 and Table 5. To answer (Q4), we observe that it is possible to perform unlearning even without finetuning to retain performance on the forgotten class. However, we also make the observation that it is possible to recover the accuracy of a forgotten class by updating the batch norm statistics by using *only samples from the remaining class*. We call this phenomenon the **BN Recall**. Thus, answering (Q5), it is necessary to modify the stored statistics in BN layers to truly forget class information.

## 7.3 DISCUSSION OF EMPIRICAL RESULTS.

In this section, we empirically answer questions (Q1) to (Q5). With (Q1), we show that the for each layer of a network, there exists a small set of High-Fidelity channels that contribute to the performance of the network. To answer (Q2), we conclude that CoBRA-P, for most cases, leads to a better sparsity vs. accuracy tradeoff against baseline data-free algorithms by at least 50% larger reduction in FLOPs. We also find, to answer (Q3), that BNFix sometimes results in better performance as compared to fine-tuning when using synthetic samples. However, BNFix is always better than no-finetuning. With reference to (Q4), we find that it is possible to perform unlearning even without finetuning to retain performance on the forgotten class. In trying to answer (Q5), we observe that when only remain class samples are for BNFix, it causes a significant increase in forget class performance.

## 8 CONCLUSION

In this paper, we study model editing in the setting where both training data and loss functions are not available, a setting not studied before. Our main contributions are algorithms devised through correlation analysis of Hifi-components- introduced for the first time here- for both Pruning Complex networks and Class Forgetting. We highlight the importance of BatchNorm statistics, which when updated, yields predictions which can be as good as those obtained from a retrained network. We provide both empirical evidence as well as a formal explaination. The results obtained here, specially those related to identifying Hi-fi components, can open doors to new research avenues useful for understanding Deep Networks. One direction for future work is to use different measures of similarity between distributions, including moment matching, Wasserstein distances, and other divergences.

**Limitations:** The techniques proposed in this work are effective when the number of classes is less than the width of the network. This may be especially true for unlearning tasks, which implicitly requires that each class is learned by disjoint set of filters.

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

APPENDIX

This appendix is organised as follows:

1. Appendix A contains details of related work
2. Appendix B contains additional experimental details
3. Appendix C contains details about BatchNorm
4. Appendix D contains derivations and proofs not presented in the main body.

## A RELATED WORK

### A.1 MODEL EDITING

In this subsection, we discuss **model editing**, which refers to techniques by which model parameters are perturbed in order to change or influence the statistical performance of the model. A variety of tasks fall under this umbrella, including pruning, model unlearning Shah et al. (2024), debiasing Santurkar et al. (2021), and continual learning Sahoo et al. (2024).

#### A.1.1 EDITING CLASSIFIERS

Interpreting and editing classifier models is an active area of research, motivated by problems such as subclass stratification (wherein subgroups within classes of a dataset can exhibit significantly different statistical performance)Sohoni et al. (2020) and debiasing Santurkar et al. (2021); Jain et al. (2022); Shah et al. (2024). The methods proposed in the latter works are of particular interest. In Jain et al. (2022), CLIP embeddings are used to find "failure directions" between samples upon which the model succeeds and those on which the model fails using an SVM; these "directions" are then used to design a variety of interventions in the weight space. In Santurkar et al. (2021), classifier prediction rules are edited by using learned rank-1 updates on a subset of layers of a DNN. Most pertinently, in Shah et al. (2024), an exhaustive approach to component attribution is used, and a variety of tasks including classwise unlearning, debiasing, editing individual predictions, and improving subpopulation robustness.

In the sequel, we discuss other methods that show that model unlearning can also be achieved via model editing.

#### A.1.2 EDITING OTHER MODELS

Model editing, while of interest to classifier models, has gained more interest in generative modeling. For instance, component editing and pruning have been successfully applied to model editing tasks in GANsLi et al. (2024); Seo et al. (2024) and diffusion models Yang et al. (2024), particularly for unlearning tasks.

### A.2 MACHINE UNLEARNING

In this subsection, we provide a detailed literature survey on machine unlearning, both with and without model editing. Machine unlearning assumes that a model $f(\cdot)$ is given, trained on a dataset $\mathcal{D}$. The dataset is then partitioned into $\mathcal{D}_r$ (i.e. the *retain* or *remember* set) and $\mathcal{D}_f$ (the *forget* set). The goal of machine unlearning is to minimize the accuracy on $\mathcal{D}_f$ while maintaining the accuracy on $\mathcal{D}_r$.

#### A.2.1 MACHINE UNLEARNING WITHOUT MODEL EDITING

Machine unlearning has gained importance in recent years owing to data privacy and security concerns Bourtoule et al. (2021); Nguyen et al. (2022). A wide variety of works exist to address this problem. Several works aim to forget data points, even in the adaptive setting, while maintaining the accuracy of the model, such as Sekhari et al. (2021); Gupta et al. (2021); Izzo et al. (2021); Golatkar et al. (2020). The work in Sekhari et al. (2021) also provides bounds on the number of samples that a model can be allowed to forget before accuracy degradation. Machine unlearning is also a significant

area of research in the space of large language models, as noted in Kurmanji et al. (2024); Eldan & Russinovich (2023), and generative models Gandikota et al. (2023).

Another aspect of machine unlearning is selective forgetting, wherein classes, groups, or sets of samples are forgotten from the network, as described in Wang et al. (2023) and the references therein. This connects machine unlearning to the continual learning setting as well, as described in Wang et al. (2024) and the references cited there. There are a variety of approaches to selective or classwise forgetting, many of which require retraining or fine-tuning on subsets of the data. Fine-tuning, which includes methods such as Golatkar et al. (2020); Warnecke et al. (2021), requires retraining the model on $\mathcal{D}_r$, assuming that after sufficient iterations, the accuracy on $\mathcal{D}_f$ would be degraded. Other works, such as Graves et al. (2021); Thudi et al. (2022) use gradient *ascent* on the loss function with $\mathcal{D}_f$, thereby destroying the accuracy of the model on $\mathcal{D}_f$.

### A.2.2 MACHINE UNLEARNING WITH MODEL EDITING

Recent works have demonstrated the promise of model unlearning by *editing* models. In Jia et al. (2023); Sahoo et al. (2024), tools for unstructured pruning are leveraged to analyze machine unlearning on sparse models, and the impact of model sparsity on such tasks. More recently, works such as Shah et al. (2024); Kodge et al.; Wang et al. (2022) directly uses structured pruning for model unlearning, by identifying components responsible for classwise predictions and removing them.

### A.3 STRUCTURED PRUNING

Structured pruning is a popular technique for improving real-world performance of models - in terms of metrics such as inference time, power consumption, and throughput - without requiring additional specialized hardware or software Hoefler et al. (2021); Blalock et al. (2020). Unlike unstructured pruning (see Frankle & Carbin (2018); Frankle et al. (2021) and the references therein for a more detailed discussion), wherein individual weights are removed, structured pruning directly reduces the number of matrix-matrix multiplications, thereby improving performance Hoefler et al. (2021). Early work on structured pruning involved pruning neurons in feedforward networks, such as LeCun et al. (1989); Hassibi & Stork (1992). More recent work typically utilizes derivatives of the loss function, such as Molchanov et al. (2019a;b); Shen et al. (2022); Li et al. (2020), which use gradients, or Hessian Liu et al. (2021a); Yu et al. (2022); Wang et al. (2020a). More recently, Lin et al. (2022) proposes estimating class-conditional gradient based saliency scores for identifying filters responsible for class-wise or group-wise predictions, with a view toward fair pruning.

### A.3.1 STRUCTURED PRUNING IN THE DATA-FREE REGIME

The space of pruning without access to the training data or loss function remains an under-researched area. There are a variety of methods that do not use training data to generate saliency scores for filters, such as Yu et al. (2018), which uses an L1 reconstruction error bound, Lin et al. (2020) which uses the rank of feature maps, ,Li et al. (2017) which uses weight norms, and Joo et al. (2021) which uses linear combinations of filters to replace redundant filters. These methods do not directly apply them to pruning in the data-free regime. In this work, we assume access to the training data distributions, with which we derive derivative-free meausres of importance of filters based on correlations between the input contributions they generate.

## B ADDITIONAL EXPERIMENTS

### B.1 SYNTHETIC DATASETS

### B.1.1 CIFAR5M

For experiments with the CIFAR10 dataset, we use CIFAR5M, a dataset containing 6 million synthetic CIFAR-10-like images sampled from a Diffusion model and labelled by a Big-Transfer model(Nakkiran et al., 2021), which we randomly sample 10,000 samples from each of the 10 classes to create our dataset. This dataset has an FID(Heusel et al., 2017) of 15.95 with respect to the CIFAR10 training set. This dataset is obtained from `https://github.com/preetum/cifar5m`.

| Dataset | Model Architecture | Original | | $\sigma$ | +Noising | | +BNFix | |
|---|---|---|---|---|---|---|---|---|
| | | Loss | Acc.(%) | | Loss | Acc.(%) | Loss | Acc.(%) |
| CIFAR-10 | ResNet-50 | 0.21 | 94.99 | 0.010 | 2.2 | 32.31 | 0.5 | 87.16 |
| | | | | 0.012 | 4.96 | 10.67 | 1.12 | 72.91 |
| | | | | 0.014 | 20.49 | 9.89 | 1.87 | 37.07 |
| | VGG19 | 0.31 | 93.50 | 0.010 | 6.04 | 18.75 | 0.5 | 86.33 |
| | | | | 0.012 | 15.11 | 11.62 | 1.23 | 59.52 |
| | | | | 0.014 | 69.69 | 10.05 | 2.01 | 26.20 |
| CIFAR-100 | ResNet-50 | 0.9 | 78.85 | 0.010 | 3.00 | 30.31 | 1.61 | 64.06 |
| | | | | 0.012 | 4.52 | 2.84 | 2.42 | 51.14 |
| | | | | 0.014 | 5.31 | 0.97 | 3.36 | 31.35 |
| | VGG19 | 1.46 | 72.02 | 0.010 | 1.62 | 62.74 | 1.55 | 66.02 |
| | | | | 0.012 | 2.27 | 48.94 | 1.62 | 62.71 |
| | | | | 0.014 | 3.75 | 13.58 | 1.80 | 58.21 |
| ImageNet | ResNet-50 | 0.96 | 76.15 | 0.010 | 4.38 | 20.56 | 1.73 | 63.63 |

**Table 3:** Effect of BNFix on noising a network. $\sigma$ represents the variance of the noise added to the network

### B.1.2   CIFAR100-DDPM

For experiements with the CIFAR100 dataset, we use CIFAR100-DDPM(Gowal et al., 2024), which we randomly downsample to contain 1,000 samples from each of the 100 classes. This dataset has an FID of 4.74 with respect to the CIFAR100 training set. We randomly sample 1,000 samples from each of the 100 classes to create our dataset. This dataset is obtained from `https://github.com/google-deepmind/deepmind-research/tree/master/adversarial_robustness/iclrw2021doing`.

### B.2   BATCHNORM NOISING

To illustrate the effect of BNFix, we will first consider an artificial editing task we call model noising. Although not a practical procedure, it serves to illustrate the effect of BNFix. The model is "edited" by adding gaussian noise to all of the learned parameters of the network. We add a zero mean random value to every learned parameter(including biases) of the model and apply BNFix for 5 iterations over the synthetic set. Table 3 showcases the performance of the model before and after noising in terms of the accuracy of the model and the value of the crossentropy loss averaged over the test set. Noising causes a dramatic fall in accuracy and increase in loss but BNFix is able to recover from around 10% to 60% of the validation accuracy across models and datasets.

### B.3   EFFECT OF NUMBER OF SAMPLES FOR BNFIX

To understand the number of samples required for BNFix, we use random pruning to prune a ResNet50 model trained on the CIFAR10 dataset to achieve 2x FLOP reduction. We then apply BNFix using the synthetic set. In Figure 3, we showcase the effect of the size of the synthetic set use and show a 95% confidence interval over 4 runs with different random subsets. We see that after around 1500 samples the gains due to adding additional samples diminish.

### B.4   TRAINING PROCEDURE

**Pretraining procedure:**   For CIFAR10 and CIFAR100, we train models using SGD Optimizer with a momentum factor of 0.9 and weight decay of $5 \times 10^{-4}$ for 200 epochs using Cosine Annealing step sizes with an initial learning rate of 0.1.

**ImageNet post training:**   For ImageNet, we use off-the-shelf pretrained models from Torchvision(Paszke et al., 2019). We train the model for 3 epochs after each iteration of CoBRA-P with learning rates of 0.1, 0.01, 0.001. After the pruning ends, we finally prune the network for 200 with a

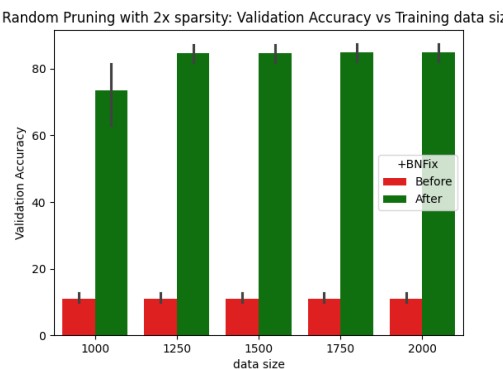

**Figure 3:** BNFix applied to a ResNet-50 model trained on CIFAR10 pruned using random channel pruning to achieve 2x FLOP sparsity.

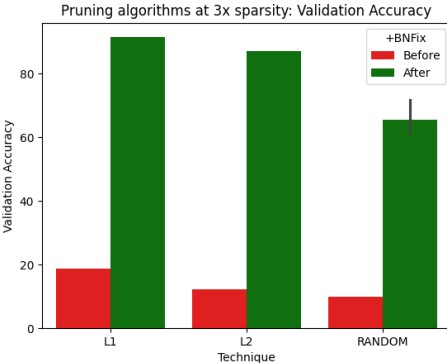

**Figure 4:** BNFix applied to a ResNet-50 model trained on CIFAR10 pruned using different pruning strategies to achieve 3x FLOP sparsity. For random pruning, we display the mean and 95% confidence interval computed over 4 runs.

batch size of 512. We use the SGD Optimizer with a momentum factor of 0.9 and weight decay of $1 \times 10^{-4}$ and Cosine Annealed step sizes with an initial learning rate of 0.1.

$L_2$ **Post training procedure:** For the synthetic training experiments mentioned in Section 7, we first prune the model using $L_2$ norm as the grouped saliency to a similar sparsity as CoBRA-P. We then train the model using 50000 samples from the synthetic dataset for 100 epochs with a batch size of 128 using SGD optimizer with momentum factor of 0.9 with inital learning rate of 0.01 and a MultiStepLR learning rate scheduler with milestones at 60 and 80 epochs.

### B.5 BNFIX AND PRUNING

We use pruning algorithms like $L_1$, $L_2$, and Random pruning on CIFAR10 trained ResNet-50 models to obtain models with 3x FLOP reduction. We then apply BNFix with 5000 synthetic samples for 20 iterations. Figure 4 shows the effectiveness of BNFix on these models, recovering upto 65% validation accuracy for this model.

## B.6    ADDITIONAL PRUNING EXPERIMENTS

| Dataset | Model | Algorithm | Acc.(%) | RF | RP |
|---------|-------|-----------|---------|-----|-----|
| CIFAR-100 | VGG19 | Unpruned | 72.02 | 1x | 1x |
| | | DFPC | 70.10 | 1.26x | 1.50x |
| | | $L_2$ | 56.46 | 1.50x | 2.40x |
| | | $L_2$ w/ ST | **72.42** | 1.50x | **2.40x** |
| | | CoBRA-P | 70.26 | **1.51x** | 2.31x |
| CIFAR10 | ResNet-101 | Unpruned | 95.09 | 1x | 1x |
| | | DFPC | 89.80 | 1.53x | 1.84x |
| | | $L_2$ w/ ST | 90.49 | 4.20 | **5.29x** |
| | | CoBRA-P | **91.20** | **4.21x** | 4.79x |
| | VGG19 | Unpruned | 93.50 | 1x | 1x |
| | | DFPC | 90.25 | 1.46x | 2.07x |
| | | $L_2$ w/ ST | 89.23 | 2.39x | **9.19x** |
| | | CoBRA-P | **91.80** | 2.39x | 5.52x |

**Table 4:** Experiments of CoBRA-P on CIFAR100 compared with baselines. RF=Reduction in FLOPs. RP=Reduction in Parameters. ST=Synthetic training, training using synthetic samples.

## B.7    ADDITIONAL FORGETTING EXPERIMENTS

We report additional experiments on class unlearning on different architectures. For VGG-19 networks, we remove the HiFi channels for the forget class of the last 12 convolution layers.

| Model | Algorithm | Forget Acc.(%) | Remain Acc. | params. removed |
|-------|-----------|----------------|-------------|-----------------|
| VGG19 | - | 93.50 | 93.50 | - |
| | CoBRA-U(0.001)(no BNFix) | 0.86 | 77.85 | 0.79M |
| | CoBRA-U(0.001) | 45.87 | 91.31 | 0.79M |
| | CoBRA-U(p=0.2) | 5.63 | 84.34 | 3.18M |

**Table 5:** Class forgetting on CIFAR10 for VGG19. CoBRA-U($p$) indicates the hyperparameter for Algorithm 3.

## B.8    CLASS UNLEARNING FOR VISION TRANSFORMERS

In this subsection, we describe how CoBRA-U can be applied to Vision Transformers to perform gradient free class unlearning without training data or access to the loss function. We focus on the SwinTransformer(Liu et al., 2021b) architecture and prune linear layers in the network. We use the distributional measure described in 4 to measure the importance of weights in linear layers of the network for the forget class. We use this measure in the form of an unstructured saliency to prune the weights of linear layers which include the $W_Q, W_K, W_V$ and MLP layers in the network. For sequence models like transformers, we compute the expectation described in equation 3 over all elements in the sequence.

We report class forgetting results on the SwinTransformer(Liu et al., 2021b) architecture trained on CIFAR-10. We train the model on the CIFAR10 dataset for 300 epochs from scratch[3] to achieve a validation accuracy of 92.31%. We apply CoBRA-U on the linear layers in a vision transformer.

---

[3]`https://github.com/jordandeklerk/SwinViT`

| Class | Forget Acc. | Remain Acc. |
|---|---|---|
| Best | 7.80% | 48.29% |
| Average | 40.52% | 60.50% |
| Worst | 90.40% | 90.74% |

**Table 6:** Training free class forgetting on CIFAR10 for SwinTransformer using CoBRA-U. Metrics are reported for the best, worst, and average over all 10 classes.

### B.9 VARIANTS OF BNFIX

Based on our analysis in Theorem 1, we develop two additional algorithms. Algorithm 4 is a variant of BNFix where the stored statistics of only channels whose coefficients in equation 6 are greater than or equal to one are updated. This ensures that only large terms of the bound are reduced by the update.

---

**Algorithm 4:** BNFix-Scale

**Input :** Batch Norm Layer $l$ with $m$ channels, dataset $\mathcal{D} = \{X_i\}_{i=1}^N$

**for** $c \in [m]$ **do**

$\quad \mu_c^{(p)} \leftarrow \frac{1}{N} \sum_{b=1}^N Y_c^l(X_b)$;

$\quad \sigma^2{}_c^{(p)} \leftarrow \sum_{b=1}^N \frac{(Y_c^l(X_b) - \mu_c^{(p)})^2}{N-1}$;

$\quad a_c = \frac{(\sigma_i^{(p)})^2 + (\mu_i - \mu_i^{(p)})^2)}{\sigma_i^2}$;

$\quad$ **if** $a_c \geq 1$ **then**

$\quad\quad \mu_c^l \leftarrow \frac{1}{N} \sum_{b=1}^N Y_c^l(X_b)$;

$\quad\quad \sigma^2{}_c^l \leftarrow \sum_{b=1}^N \frac{(Y_c^l(X_b) - \mu_c^l)^2}{N-1}$;

---

We compare the performance of these variants as a substitution for retraining for CoBRA-P for a VGG model pretrained on CIFAR10.

### B.10 HYPERPARAMETERS FOR EXPERIMENTS

We randomly sample 2000 data points from distributional access for computing HiFi channels and for BNFix. For CoBRA-P, we typically set $p = 0.05$. We perform BNFix for 10 epochs as a substitution for retraining. We use 2000 samples from a synthetic dataset for BNFix. For ImageNet, we use 20000 samples from the unlabelled imagenet test set.

### B.11 DETAILED COBRA ALGORITHMS

In this section we provide details of CoBRA-P and CoBRA-U.

| Algorithm | Acc. | RF | RP |
|---|---|---|---|
| - | 93.50 | 1x | 1x |
| No BNFix | 63.75 | 2.42x | 4.72x |
| BNFix-Scale | 89.10 | 1.97x | 4.18x |
| BNFix | 92.47 | 1.91x | 3.98x |

**Table 7:** Performance of variants of BNFix as a substitution for retraining for CoBRA-P on a VGG19 model trained on CIFAR10.

**Algorithm 5:** CoBRA-P

**Input:** Model, keepRatio $p$, Samples $\mathcal{D} = \{(X_i, y_i)\}_{i=1}^N$
**Output:** Edited model
**Function** `Prune`(*Model, p, D*):

    `// Find the set of all coupled channels`
    DFCs ← FindCoupledChannels(Model);
    **for** $CC \in DFCs$ **do**
        HiFiChannels← `ComputeHiFiSet`(*CC,p,D*);
        EditableChannels← $[C_{in}^{CC}]\backslash$HiFiChannels;
        **for** $l \in CC$ **do**
            **for** $i \in EditableChannels$ **do**
                `// Prune the input channel in each layer of the DFC`
                PruneInputChannel($l, i$);
            Compute $\delta^{(l)\star}$ based on equation 8 using $\mathcal{D}$;
            **for** $i \in HiFiChannels$ **do**
                InputChannel($l, i$) ← $\delta^{(l)\star}\cdot$ InputChannel($l, i$);
    `// Run Algorithm 2 for all BatchNorm layers in the model`
    **for** $l_{bn} \in FindBNLayers(Model)$ **do**
        BNFix($l_{bn}, \mathcal{D}$);

**Result:** Model

---

**Algorithm 6:** CoBRA-U

**Input:** Model, keepRatio $p$, Forget Samples $\mathcal{D}_f$, Remain samples $\mathcal{D}_r$
**Output:** Edited model
**Function** `Unlearn`(*Model, p, $\mathcal{D}_f$, $\mathcal{D}_r$*):

    `// Find the set of all coupled channels`
    DFCs ← FindCoupledChannels(Model);
    **for** $CC \in DFCs$ **do**
        **for** $l \in CC$ **do**
            HiFiChannels← `ComputeHiFiSet`(*l,p,Df*);
            EditableChannels←HiFiChannels;
            **for** $i \in EditableChannels$ **do**
                `// Prune the input channel in each layer of the DFC`
                PruneInputChannel($l, i$);
            Compute $\delta^{(l)\star}$ based on equation 8 using $\mathcal{D}_r$;
            **for** $i \in HiFiChannels$ **do**
                InputChannel($l, i$) ← $\delta^{(l)\star}\cdot$ InputChannel($l, i$);
    `// Run Algorithm 2 for all BatchNorm layers in the model`
    **for** $l_{bn} \in FindBNLayers(Model)$ **do**
        BNFix($l_{bn}, \mathcal{D}_r$);

**Result:** Model

### B.12 VALIDATING HiFi HYPOTHESIS

In this section, we present the plots for the fidelity score computed as per 7.

## C ADDITIONAL DETAILS ABOUT BATCHNORM

### C.1 BATCHNORM2D

For multi-channel data like images, BatchNorm is modified "to satisfy the convolution property"(Ioffe & Szegedy, 2015). Let $\Phi^l(x)$ denote the input to the $l^{th}$ layer of a neural network with $L$ layers on input $x$. Let the $l^{th}$ layer be a Convolution layer with $m$ ouput channels, for a single multi channel sample $x$, let $Y^l(x) \in \mathbb{R}^{m \times d}$ be the flattened representation of the output which is computed according to equation 1. The output of the BatchNorm layer(called BatchNorm2D in the multi-

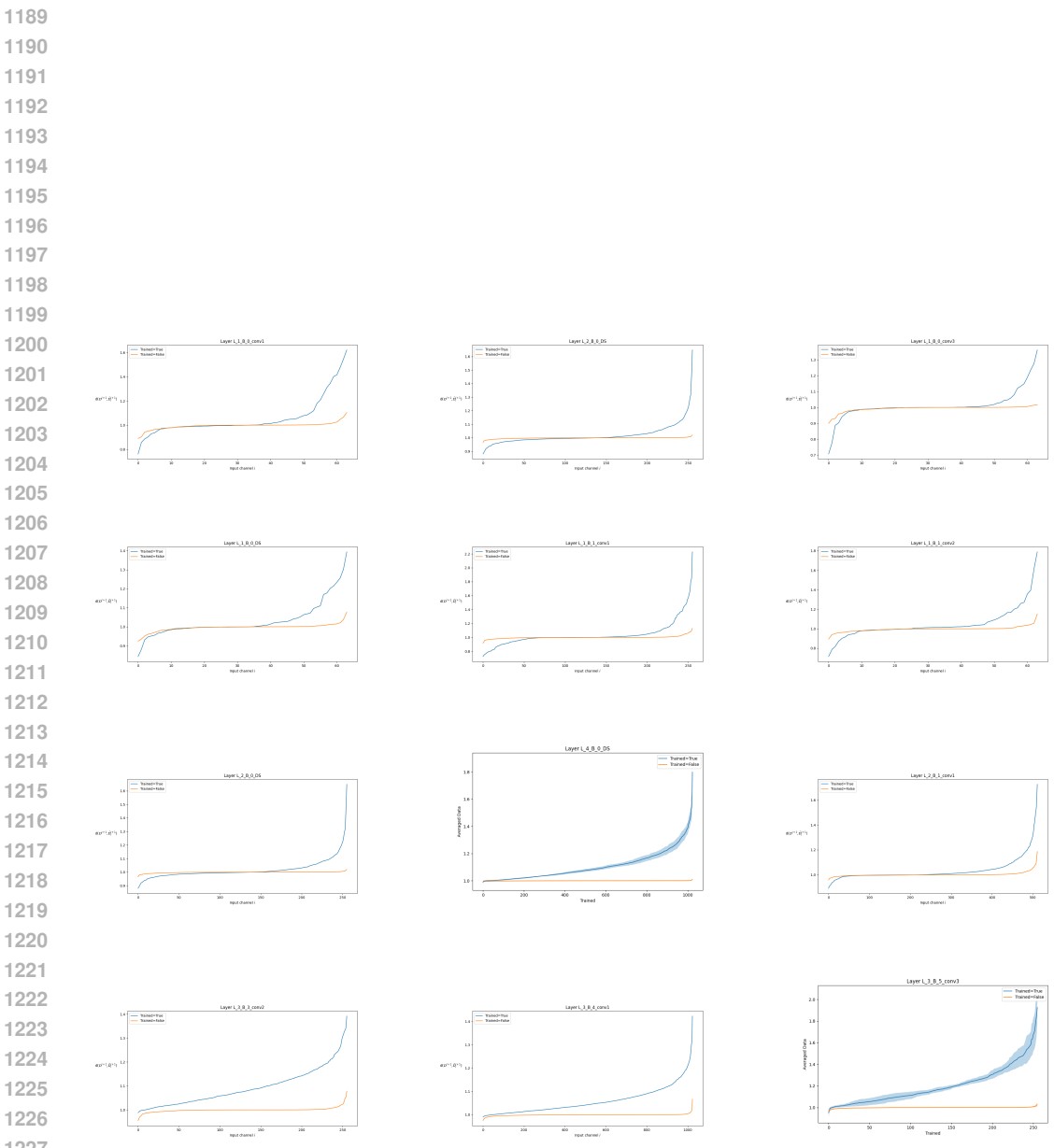

**Figure 5:** Comparison of distributional similarity between input contributions and output feature map.

channel case), $V(X) = BN_{\gamma,\beta}^{l+1}(X) \in \mathbb{R}^{m \times d}$, is given by

$$V_c(X) = \gamma_c Z_c(X) + \beta_c \mathbb{1} \quad \text{where} \quad Z_c(X) = \frac{Y_c^l(X) - \mu_c \mathbb{1}}{\sigma_c} \quad \forall c \in [m] \tag{9}$$

Where $\mathbb{1}$ is a vector of ones, $\mu_c = \frac{1}{N} \sum_{i=1}^N \frac{Y_c^l(x_i)^\top \mathbb{1}}{d} \approx \mathbb{E}_X[\frac{Y_c^l(X)^\top \mathbb{1}}{d}]$ and $\sigma_c^2 \approx \text{Var}_X(\frac{Y_c^l(X)^\top \mathbb{1}}{d})$ are stored data statistics and $\gamma \in \mathbb{R}^m$ and $\beta \in \mathbb{R}^m$ are the learned scale and shift parameters that determine the first two moments of the output of the layer, i.e., for the random variable $\hat{V}(X) = \mathbb{1}^\top V_c(X)/d$, the moments are $\mathbb{E}_X[\hat{V}(X)] = \beta_c$ and $\text{Var}_X(\hat{V}(X)) = \gamma_c^2$.

## C.2 BATCHNORM DURING TRAINING

We describe the behavior of BatchNorm1D during training, which is similar to that of BatchNorm2D. For an input batch of size B, let the output of the linear layer, the $l^{th}$ layer in the network, be $\phi^l(X) \in \mathcal{R}^{B \times d}$. Then, the output is given by,

$$Y_c(x_i) = \gamma_c Z_c(x_i) + \beta_c \quad \text{where} \quad Z_c(x_i) = \frac{\phi^l(x_i) - \frac{1}{B} \sum_{b=1}^B \phi_c^l(x_b)}{\sqrt{\frac{1}{B} \sum_{j=1}^B \left( \phi_c^l(x_j) - \frac{1}{B} \sum_{b=1}^B \phi_c^l(x_b) \right)^2}} \tag{10}$$

The estimate over the whole training set from equation 2 is now replaced with batch estimates. Observe that $Z$ are being normalized and, in an average sense, represent zero mean unit variance random variables. To compute the stored statistics to use during inference, at every forward pass during training, a running estimate of the mean and variance are stored in the layer. This running estimate is used in equation 2.

# D PROOFS

In this section, we provide proofs for the main theoretical results proposed in this work. Specifically, we propos

## D.1 PROOF OF LEMMA 1

**Lemma 1** (Loss of a well trained model expressed with BatchNorm). *Consider a model that satisfies assumptions 5. We can express an upper bound on the expected loss during inference in terms of the statistics of the output of the BatchNorm layer $V(X) \in \mathbb{R}^m$.*

$$|\mathbb{E}[\mathcal{L}(V(X))] - \mathcal{L}(\beta)| \leq \frac{K}{2} ||\gamma||^2 \tag{5}$$

*Proof.* From fact 1,

$$L(Y(X)) = L(\beta + GZ) = L(\beta) + \nabla L(\beta)^\top GZ + \frac{1}{2} Z(X)^\top GH(\beta, GZ(X)) GZ(X)$$

The proof follows from the assumptions on the Hessian. **Comments:** $|E(X)| \leq E(|X|)$. $\qquad \square$

## D.2 PROOF OF THEOREM 1

**Theorem 1.** *Let the $l^{th}$ layer of a network be a BatchNorm layer as described in 2 with stored data statistics $\mu_c$ and $\sigma_c^2$. Editing components of preceeding layers causes a change in the distribution of the intermediate representation to some $Y^{(p)}(X)$, with modified moments $\mu^{(p)}$ and $(\sigma^{(p)})^2$. The output of BatchNorm after editing is then, $\mathbf{V}^{(p)} = GZ^{(p)} + \beta$ where $Z^{(p)} = \frac{Y_c^{(p)}(X) - \mu_c}{\sigma_c}$. Then,*

$$|\mathbb{E}[\mathcal{L}(\mathbf{V}^{(p)}(X))] - \mathcal{L}(\beta)| \leq \frac{K}{2} \left( \sum_{i=1}^d \gamma_i^2 \left( \left( \frac{\sigma_i^{(p)}}{\sigma_i} \right)^2 + \left( \frac{\mu_i^{(p)} - \mu_i}{\sigma_i} \right)^2 \right) \right) \tag{6}$$

*Proof.* Let $\mathbf{V}^{(p)}$ be the output at the batchnorm layer for the edited model with $E(\mathbf{V}^{(p)}) = \mu^{(p)}, E((\mathbf{V}_i^{(p)})^2) = \left(\mu_i^{(p)}\right)^2 + \left(\sigma_i^{(p)}\right)^2, \forall i \in [d].$

Define $U_i = \frac{\mathbf{V}_i^{(p)} - \mu_i}{\sigma_i}$. Consider $\widehat{\mathbf{V}^{(p)}} = GU + \beta$

$$|\mathbb{E}(L(\widehat{\mathbf{V}^{(p)}})) - L(\beta)| \le \nabla L(\beta)^\top U + \mathbb{E}\frac{1}{2}\|H(\beta, GU)\|_2\|GU\|_2^2 \tag{11}$$

$$\le \sum_{i=1}^{d} g_i \gamma_i \frac{\mu_i^{(p)} - \mu_i}{\sigma_i} + \frac{K}{2}\left(\sum_{i=1}^{d}\gamma_i^2\left(\left(\frac{\sigma_i^{(p)}}{\sigma_i}\right)^2 + \left(\frac{(\mu_i^{(p)} - \mu_i)}{\sigma_i}\right)^2\right)\right) \tag{12}$$

where $g_i = \nabla_i \mathcal{L}$. Applying assumption 5 finishes the proof.

$\square$

### D.3 PROOF OF THEOREM 2

**Theorem 2.** *Let $s^l \in \{0,1\}^{C_{in}} = [\mathbf{1}_K; \mathbf{0}_{C_{in}-K}]$, where $\mathbf{1}_K$ is a vector of $K$ ones, and $\mathbf{0}_{C_{in}-K}$ is a vector of $C_{in} - K$ zeros; we ignore the subscripts for brevity in the sequel. Define $\delta_c \in \mathbb{R}^{C_{in}}$ such that $\delta_{ci} = 0$ when $s_i = 0$. We solve $\hat{W}_{ci}^{l+1} = \hat{\delta}_{ci}^{l+1} W_{ci}^{l+1}$, where $\bar{\delta}_{ci}^{l+1} = [\hat{\delta}_{ci}^{l+1}; \mathbf{0}_{C_{in}-K}]$ that satisfies*

$$\hat{\delta}_c^{l+1} = \underset{\delta_c \in \mathbb{R}^K}{\arg\min} \, \mathsf{RE}_c^l([\delta, 0]) = P_c^{-1}p_c \quad and \quad \frac{\mathsf{RE}_c^l(s^l) - \mathsf{RE}_c^l(\bar{\delta}_{C_{in}}^{l+1})}{\mathsf{RE}_c^l(s^l)} \le 1 - \frac{\|1 - \bar{\delta}_{ci}^{l+1}\|^2}{\kappa(Q_c^{l+1})(C_{in} - K)} \tag{8}$$

*where $\delta_c^\star$ is a vector containing the optimal values of $\delta_{ci}$, $Q_{c,ij} = \mathbb{E}\left[(W_{cj}^{l+1})^\top \Phi_j(X)^\top \Phi_i(X) W_{ci}^{l+1}\right]$, $P_{c,ij} = Q_{c,ij}$ and $p_{c,i} = \mathbb{E}\left[(Y_c^{l+1})^\top \Phi_i^l(X) W_{ci}^{l+1}\right]$ when $s_i, s_j = 1$, and $\kappa(Q_{c,ij})$ denotes the condition number of $Q_{c,ij}$.*

*Proof.* First, note that

$$\mathsf{RE}_c^{l+1}([\delta; \mathbf{0}_{C_{in}-K}]) = \mathbb{E}[\|Y_c^{l+1}(X) - \sum_{i:s_i=0}\delta_i \Phi_i^l(X) W_{ci}^{l+1}\|^2]$$

$$= \mathbb{E}[\|\sum_i \delta_i \Phi_i^l(X) W_{ci}^{l+1} - \sum_{i:s_i=0}\delta_i \Phi_i^l(X) W_{ci}^{l+1}\|^2]$$

$$= (1 - [\delta; \mathbf{0}_{C_{in}-K}])^\top Q(1 - [\delta; \mathbf{0}_{C_{in}-K}]).$$

We can rewrite this as

$$\underset{\delta}{\arg\min} \, \mathsf{RE}_c^{l+1}([\delta; \mathbf{0}_{C_{in}-K}]) = \underset{\delta}{\arg\min} \, \delta^\top P_c \delta - 2p_c^\top \delta = P_c^{-1}p_c.$$

To measure the error, note that

$$\mathsf{RE}_c^{l+1}(s^l) = (1 - s^l)^\top Q(1 - s^l).$$

Thus, we have

$$\frac{\mathsf{RE}_c^l(s^l) - \mathsf{RE}_c^l(\bar{\delta}_c^{l+1})}{\mathsf{RE}_c^l(s^l)} = 1 - \frac{\mathsf{RE}_c^l(\bar{\delta}_{C_{in}}^{l+1})}{\mathsf{RE}_c^l(s^l)} = 1 - \frac{(1 - \bar{\delta}_c^{l+1})^\top Q(1 - \bar{\delta}_c^{l+1})}{(1 - s^l)^\top Q(1 - s^l)}$$

$$\le 1 - \frac{\lambda_{\min}(Q)\|1 - \bar{\delta}_c^{l+1}\|^2}{\lambda_{\max}(Q)(C_{in} - K)} = 1 - \frac{\|1 - \bar{\delta}_c^{l+1}\|^2}{\kappa(Q)(C_{in} - K)}.$$

$\square$

# E   RECONSTRUCTION ERROR AND HIFI COMPONENTS

**Reconstruction Error and the Fidelity Score**   A useful way to identify HiFi components is by measuring the expected reconstruction error, as fidelity implies reconstructability. We define reconstion error as follows. First, let $s \in \{0,1\}^{C_{in}}$ satisfy $s_i = 1$ if $i \in$ keep and $0$ otherwise, and let $\hat{Y}^l(X) = \sum_i s_i A_i^l(X)$. The expected *local reconstruction error* at layer $l$ between $Y^l(X)$ and $\hat{Y}^l(X)$, as well the reconstruction error of a single output channel, are

$$\mathsf{RE}^l(s) = \mathbb{E}_X \left[ \|Y^l(X) - \hat{Y}^l(X;s)\|^2 \right]$$

$$\mathsf{RE}_c^l(s) = \mathbb{E}_X \left[ \|Y_c^l(X) - \hat{Y}_c^l(X;s)\|^2 \right] \quad \text{and} \quad \mathsf{RE}^l(s) = \sum_c \mathsf{RE}_c^l(s). \tag{13}$$

Next, we consider the reconstruction error after a 1D BatchnNorm layer, assuming that the stored statistics are not updated after editing.

$$\mathsf{RE}_c^{l+1}(s) = \frac{1}{2}\mathbb{E}_X \left[ \left( V_c(X) - \hat{V}_c(X;s) \right)^2 \right] = \frac{\gamma_c^2}{2} \left( 1 + \frac{\hat{\sigma}_c^2}{\sigma_c^2} - \frac{(\hat{\mu}_c - \mu_c)^2}{\sigma_c^2} - 2\frac{\hat{\sigma}}{\sigma_c}\rho_c(s) \right) \tag{14}$$

where $V_c(X)$, $\hat{V}_c(X)$, $\mu_c$, $\sigma_c$, $\hat{\mu}_c$, and $\hat{\sigma}_c$ are defined as in equation 2 .
When we adjust the batchnorm statistics,, the reconstruction error is given by,

$$\mathsf{RE}_c^{l+1}(s) = \gamma_c^2(1 - \rho_c(s)) \tag{RE-BN}$$

This adjustment shows that the reconstruction error is a function of the *correlation* between the input contributions and the output. This motivates us to define $\mathsf{FS}(i)$ using the centered distributions. In Section 5, we rigorously analyze the effect of adjusting the BatchNorm statistics on the loss of the network.

**Deriving BNFix From Reconstruction Error**   Consider the output of the BatchNorm layer before and after pruning where the stored statistics are changed after pruning. Let $V(X) = BN_{\gamma,\beta}(Y(X), \mu, \sigma)$ and $\hat{V}(X; s, \mu', \sigma') = BN_{\gamma,\beta}(\hat{Y}(X), \mu', \sigma')$, where $\mathbb{E}[Y_c(X)] = \mu_c$, $\mathrm{Var}(Y_c(X)) = \sigma_c^2$, $\mathbb{E}[\hat{Y}_c(X;s)] = \hat{\mu}_c$ and $\mathrm{Var}(\hat{Y}_c(X;s)) = \hat{\sigma}_c^2$. The reconstruction error for output channel $c$ is given by,

$$\mathsf{RE}_c^{l+1}(s) = \frac{1}{2}\mathbb{E}_X \left[ \left( V_c(X) - \hat{V}_c(X;s,\mu',\sigma') \right)^2 \right] = \frac{\gamma_c^2}{2} \left( 1 + \frac{\hat{\sigma}_c^2}{\sigma_c'^2} - \frac{(\hat{\mu}_c - \mu_c')^2}{\sigma_c'^2} - 2\frac{\hat{\sigma}}{\sigma_c'}\rho_c(s) \right)$$
$$\tag{15}$$

where $\rho_c(s) = \frac{\mathrm{Cov}(Y_c(X), \hat{Y}_c(X;s))}{\sigma_c\sqrt{\mathrm{Var}(\hat{Y}_c(X;s))}}$. When $\mu' = \hat{\mu}$ and $\sigma' = \hat{\sigma}$, the reconstruction error is given by,

$$\mathsf{RE}_c^{l+1}(s) = \gamma_c^2(1 - \rho_c(s)) \tag{RE-BN}$$

**Reconstruction Error for Structured Pruning**   Solving equation $\mathsf{Prune}$ without access to the training data or loss function can now be formulated as *minimizing the reconstruction error between the edited feature map and the original*. Thus, we formulate the problem of structured pruning with a fixed budget as follows. For a layer with $C_{in}$ filters, and a sparsity budget of $B$ filters, we write

$$s^* = \operatorname*{arg\,min}_{s \in \{0,1\}^{C_{in}}} (\mathbb{1} - s)^T Q_c (\mathbb{1} - s) \quad \text{s.t.} \quad \sum_i s_i \le B \tag{16}$$

where $Q_c$ is a symmetric matrix with elements $Q_{cij} = W_{ic}^\top \mathbb{E}_X \left[ \Phi_i(X)^\top \Phi_j(X) \right] W_{jc}$.

The optimization problem in 16 is a binary optimization problem and thus NP-Hard , and can be reduced to a graph problem like maximum independent set by considering that $Q$ represents the adjacency matrix of a graph. Solving equation 16 provides an independent set of size $B$. Based on this observation, we use a simple heuristic, analogous to the degree of each vertex, to find solutions to equation 16. Consider minimizing the reconstruction error for a single output channel, say $c$. With

our graph analogy, we remove vertices with the lowest degree, corresponding to channels with the lowest row sums of the matrix $Q_c$. The row sum corresponding to each channel, $R_{ci}$ is

$$R_{ci} = \sum_j Q_{cij} = \sum_j W_{ic}^\top \mathbb{E}_X \left[ \Phi_i(X)^\top \Phi_j(X) \right] W_{jc} = W_{ic}^\top \mathbb{E}_X \left[ \Phi_i(X)^\top Y_c(X) \right] \quad \textbf{(RowSum)}$$

Based on equation **RE-BN**, we normalize the input contribution and the output to zero mean random variables due to the presence of BatchNorm layers. This algorithm is stated formally in Algorithm 1.

**Reconstruction Error for Classwise Unlearning**   We aim to use the HiFi hypothesis to *unlearn* a class from a well-trained model, by removing a small, fixed number of filters responsible for predictions of that class. We formulate the problem of classwise unlearning via model editing using the Reconstruction Error to guide the edit. Our goal is to maximize the reconstruction error on the forget class drawn from distribution $\mathcal{D}_f$ while minimizing the reconstruction error on the remaining classes, which we denote by $\mathcal{D}_r$. Similar to equation 16, we write

$$\arg\max_{s \in \{0,1\}^{C_{in}}} (\mathbb{1} - s)^T \left( Q_c^f - \alpha Q_c^r \right) (\mathbb{1} - s) \quad \text{s.t.} \quad \sum_i s_i \geq B \qquad \textbf{(Forget)}$$

where $Q_c^f$ is a symmetric matrix with elements $Q_{cij}^f = W_{ic}^\top \mathbb{E}_{X \sim \mathcal{D}_f} \left[ \Phi_i(X)^\top \Phi_j(X) \right] W_{jc}$, $Q_c^r$ $Q_{cij}^r = W_{ic}^\top \mathbb{E}_{X \sim \mathcal{D}_r} \left[ \Phi_i(X)^\top \Phi_j(X) \right] W_{jc}$, and $\alpha$ is a hyperparameter that penalizes the reconstruction error on $\mathcal{D}_r$. Our experiments show that typically, setting $\alpha = 0$ suffices, particularly for wide networks such as ResNet50 and VGG19 trained on CIFAR10.

