# OpenReview forum: "No Training Data, No Cry: Model Editing  without Training Data or Fine-tuning"
_ICLR.cc/2025/Conference — Submitted to ICLR 2025_

### Official Review · Reviewer_d7ey · 2024-10-31

**Soundness:** 3
**Presentation:** 2
**Contribution:** 3
**Rating:** 6
**Confidence:** 3

**Summary:**

This paper deals with the finetuning-free model editing of ResNet models without accessing the original training data. The authors hypothesize that High Fidelity (HiFi) components of the model take charge of overall performance retainment and propose determining the pruning parts from a model based on the reconstruction score. The authors further provide a novel theoretical analysis of the batch normalization statistic to characterize the model performance after editing. Evaluation was performed over model pruning and class-level unlearning tasks.

**Strengths:**

* This paper provide a novel theoretical analysis on batch normalization statistics to discuss post-edited model performance

**Weaknesses:**

* **Limited applicability of the proposed method**
  * Although ResNet models are still popular in some cases, given that Vision Transformer (ViT) or other transformer-based models are dominant in many applications, the aim of this study limits its impact compared to previous work on model editing [1].
  * Could the insights provided in this work have some implications for the transformer-style models?
* **Limited validation scope**
  * Although this paper provides some theoretical insights, the empirical validation is too weak in terms of
 the number of baseline methods, datasets, and experimental settings.
  * Could more baseline methods for the unlearning task be considered? Either data-free [2] or not [3].
  * Could more datasets be considered here for the unlearning task?
* **Insufficient empirical advantage**
  * The authors claim that the proposed method achieves a good trade-off between accuracy and efficiency. However, the proposed method actually could not achieve good accuracy compared to baseline methods, and the benefits of enhanced efficiency are also not so strong on both pruning and unlearning tasks.
* **Reliance on external data (through distributional access)**
  * Although the proposed method does not use an explicit training dataset on which the mode is trained, it still requires some samples from a similar distribution. This weakens the practical usefulness of the proposed method compared with truly data-free methods such as task arithmetic-based unlearning [2]
  * Could the authors provide an ablation study for the size of the external dataset used for proposals?
* **Bad presentation quality**
  * In the introduction and experiment section, the author does not insert space between paragraphs, which makes the reading hard.
  * The quality of the figure and table is so bad in terms of font size and resolution.
  * There is incorrect labeling of assumption 5 in line 328
  * Notations are complex beyond need and somewhat unclear. One example is lines 177-178.



> Reference
1. Decomposing and Editing Predictions by Modeling Model Computation, Shah et al. 2024
2. Editing Models with Task Arithmetic, Ilharco et al. 2024
3. Decoupling the Class Label and the Target Concept in Machine Unlearning, Zhu et al. 2024

**Questions:**

See the weakness section.

---

> ### Author Response · Authors · 2024-11-25
> **Response to reviewer d7ey**
>
> We thank the reviewer for their time and finding value in our theoretical analysis of BatchNorm at inference.
>
> Based on their comments, we have made changes to the manuscript to enhance clarity and have provided additional results on unlearning for Vision Transformers.
>
> We first wish to clarify a statement in the summary of the reviewer:
> > "The authors hypothesize that High Fidelity (HiFi) components..."
>
> We wish to clarify that we are the first work to define and introduce the notion of HiFi components in Lines 228-230 in Section 4.1. We hypothesize that such channels are few in well trained models in Hypothesis 1 in lines 285-288.
>
> We now address the individual concerns raised by reviewer d7ey.
>
> > **W1: Limited applicability of the proposed method**
>
> In this paper, there are several proposed methods for model editing: RowSum Heuristic described in Lines 277-280 for identifying HiFi components; Algorithm 2 (BNFix) in Section 5 and Theorem 2(Weight compensation) in Section 6 as an alternatives to retraining in our setting; A novel analysis, Theorem 1, of BatchNorm at inference; CoBRA algorithms, Algorithm 3,  for editing models with complex interconnections without training data or the loss function.
> As noted by reviewer ZvuL, the proposed method can be applied across areas such as continual learning and explainability but is beyond the scope of this work and remains open for future explorations.
>
> > **Although ResNet models are still popular in some cases, given that Vision Transformer (ViT) ... compared to previous work on model editing [1]**
>
> The correlation based techniques and the CoBRA algorithms developed in Section 6 are **readily applicable** to transformer based models. [1] requires access to the loss function to perform component attribution and would not be applicable to the setting in this work.
>
> > **Could the insights provided in this work have some implications for the transformer-style models?**
>
> As per your suggestion, we have added Appendix B.8 to describe how to edit QKV and MLP layers in transformer based vision models in the updated manuscript along with experiments on classwise unlearning on Swin-Transformer models.
>
> >   **Limited validation scope**
>
> We provide emperical validation across multiple tasks, pruning and unlearning on CIFAR10, CIFAR100, and ImageNet in Section 7 in Tables 1 and 2. The datasets we use in our experiments match those of our state of the art baselines [5, 6], which investigates machine unlearning (specifically classwise unlearning) and structured pruning for image classification. The methods proposed in [2] require multiple pre-trained models (or the ability to train new models on desired data subsets) and would not be applicable in our setting. Technique [3] would require access to training data and the loss function, making it unsuitable for comparison in this setting.
>
> >   **Insufficient empirical advantage**
>
> We would like to highlight the empirical advantage of our work on both pruning and classwise unlearning.
>
> **Pruning**
>  On CIFAR10 and CIFAR100 datasets, our method achieves 2.8x improvement in sparsity over the nearest baseline for ResNet50 without finetuning, and outperforms L2-based pruning **with finetuning on synthetic data**, as we show in Table 1 in Section 7.
>
> **Unlearning**
> On ResNet50 trained on CIFAR10, we achieve superior Forget Class accuracy than the baselines proposed in [5] **without any fine-tuning**, whereas the baselines require multiple epochs of fine-tuning; we refer the reviewer to Table 2 in Section 7 of the manuscript for further details.
>
> > **Reliance on external data (through distributional access)**
>
>  Our method is inspired by works such as *Dreaming to Distill* [4], which motivates the use of samples drawn from a similar distribution to overcome concerns of privacy and security. The methods proposed in [2] require multiple pre-trained models (or the ability to train new models on desired data subsets) and would not be applicable in our setting.
>
> > **Could the authors provide an ablation study for the size of the external dataset used for proposals?**
>
> We refer the reviewer to Appendix B.1 for details of the external datasets and Figures 3 and 4. And Appendix B.3 for the study on the size of the synthetic dataset. We experiment with varying number of samples to conclude that 1500 synthetic samples are sufficient for BNFix and use the same number of samples for computing _RowSum_.
>
> > **Bad presentation quality**
>
> We thank the reviewer for these corrections. We have made updates to the manuscript to reflect them.
> > References
>
> 1.  Decomposing and Editing Predictions by Modeling Model Computation, Shah et al. 2024
> 2.  Editing Models with Task Arithmetic, Ilharco et al. 2024
> 3.  Decoupling the Class Label and the Target Concept in Machine Unlearning, Zhu et al. 2024
> 4.  Dreaming to Distil, Yin et al, 2021.
> 5. Model sparsity can simplify unlearning, Jia et al, 2023
> 6. DFPC, Narshana et al., 2023

---

> ### Comment · Reviewer_d7ey · 2024-11-25
> **Response to authors' rebuttals**
>
> I sincerely appreciate your thoughtful rebuttal together with additional experiments that are related to my concerns. The overall presentation was remarkably improved (although there is still room for improvement in terms of figures and tables)! Besides, the authors addressed concerns about the validation scope and reliance on external data.
>
> _Therefore, I raised my rating from 3 to 5._
>
> * I think the biggest contribution of this work is a novel theory, and I wanted to grasp the applicability of this kind of theoretical analysis to the broader setting where the model does not use batch normalization (non-BatchNorm architectures are very common for modern neural networks).
> * Although the authors provide an experiment with Swin-Transformer and I appreciate it, there is no theoretical implication for non-BN architecture.
> * Also, I still wouldn't be so sure about the significance of improvement. Although the authors claim this, the trade-offs between accuracy and efficiency (Tables 1 and 2) are not so significantly improved compared with the baseline to my eyes.
>
> Anyway, I am open to further discussion upon the extended deadline.
>
> Thank you,
>
> Reviewer d7ey

---

> > ### Author Response · Authors · 2024-11-26
> > **Follow-up discussion with Reviewer d7ey**
> >
> > Thank you for your prompt reply and for raising your score!
> > We would be happy to engage with the reviewer to clarify their concerns with our work.
> >
> > >  **I think the biggest contribution of this work is a novel theory, and I wanted to grasp the applicability of this kind of theoretical analysis to the broader setting where the model does not use batch normalization (non-BatchNorm architectures are very common for modern neural networks).**
> >
> >  - We point out that our analysis proposed in Section 5 applies only to BatchNorm. BatchNorm is **the only normalization techniques that rely on stored statistics ($\mu$, $\sigma$)**, among popular normalization techniques such as LayerNorm or RMSNorm.
> >  - Moreover, our theoretical analysis is not limited to just rectifying normalization techniques - we would like to point the reviewer to Section 4, which proposes a principled mechanism for identifying components responsible for a model's predictions, with potential applications beyond classwise unlearning and structured pruning, as noted by Reviewer ZvuL [here](https://openreview.net/forum?id=wLR9d5ZFpY&noteId=IiI4u8NOm4).
> >
> > >   **Although the authors provide an experiment with Swin-Transformer and I appreciate it, there is no theoretical implication for non-BN architecture.**
> > - Our experiment on unlearning in the Swin-Transformer (presented in Appendix B.8) showcases that our **distributional approach** can be applied to pruning transformers as well. The DIS score (Equation 3) in the main manuscript is applicable to ViTs as well, showing that the analyses proposed in Sections 4 and 6 **are applicable for non-BN architectures as well**.
> >
> > >   **Also, I still wouldn't be so sure about the significance of improvement. Although the authors claim this, the trade-offs between accuracy and efficiency (Tables 1 and 2) are not so significantly improved compared with the baseline to my eyes.**
> >
> > - In the experiments presented in Tables 1 and 2, note that we present results in the setting **without the training data and the loss function** (with the exception of the Imagenet pruning experiments). Our results in Table 1, despite **being compared to results  with fine-tuning** (i.e. L2-ST, which incorporates fine-tuning on synthetic data), and still performs comparably with current state of the art. This showcases the effectiveness of the CoBRA-P algorithm.
> > - Moreover, in Table 2, the results baselined against (Jia et al, 2023), **requires 10 epochs of fine-tuning**. Our method is comparable to the state of the art **without any additional fine-tuning,** highlighting the effectiveness of the CoBRA-U algorithm.
> >
> > We hope to continue this discussion with the reviewer and clarify any concerns preventing the reviewer from raising their score to accept this work.

---

> ### Comment · Reviewer_d7ey · 2024-11-26
> **2nd response to authors' rebuttals**
>
> Thank you for the active discussion and further clarification.
>
> I had some misunderstandings about the applicability of your theory and the details behind the experiment.
>
> * Authors' additional responses addressed my remaining concerns in terms of the impact of theory and practical benefits.
> * I would strongly recommend the authors 1) clearly denote the training cost comparison such as total runtime in the tables so that they highlight the training-free characteristics (which are distinctive from the baseline) and 2) insert the result for Swin Transformer within the 10-page main body of paper rather than appendix upon camera-ready revision.
>   * Also, I believe the authors will further improve the presentation quality (increasing the font size in figures/tables, adjusting margin/space, and so on) upon camera-ready if this paper is accepted.
> * Overall, although I still see some weaknesses, e.g., marginal improvement and limited scope of validation, other concerns that I had are all addressed by authors' sincere refute and I believe the paper has been significantly improved through the rebuttal period.
>
> _Based on this, I am raising my score from 5 to 6, and increased all the soundness/presentation/contribution items by one point._
>
> The reviewer d7ey sincerely appreciates the authors' heartfelt discussion.

---

> > ### Author Response · Authors · 2024-11-27
> > **Thanks to Reviewer d7ey!**
> >
> > We sincerely thank the reviewer for raising their score and for actively engaging in discussions that helped strengthen this work.
> >
> > We are pleased that we were able to address your concerns regarding the theoretical and practical impact of our work.
> >
> > If this work is accepted, we will ensure that the reviewer's suggestions regarding
> > 1. Additional results on ViT models
> > 2. Comparisons of the training time for each method
> > 3. Further improvements in the presentation of the work such as improved images and tables
> >
> > are incorporated into the main body of the manuscript under the conference page limit.
> >
> > We will also let the reviewer know of any changes made following discussions with other reviewers. We would be glad to engage in these discussions with you to address any furthur concerns.

---

### Official Review · Reviewer_ZvuL · 2024-11-03

**Soundness:** 2
**Presentation:** 1
**Contribution:** 2
**Rating:** 3
**Confidence:** 4

**Summary:**

This paper mainly focuses on the model editing task, emphasizing the setting without training data or loss functions.
To detour access to the data or loss functions, the authors investigate the 'distributional' behavior of network layer outputs, which is not a 'sample-wise' behavior. Based on the finding that a very limited number of components of networks contribute to the learned outputs (called **HiFi** components), the authors have proposed to freeze the HiFi components and adjust the batch normalization to compensate for the changes in the distributional behavior. To verify their approaches, they have provided two types of tasks, i.e., pruning and unlearning.

**Strengths:**

**Strength 1:** The main strength of this paper is that the authors' viewpoint to scrutinize the distributional behavior of networks rather than the sample-wise network sensitivity can be a key strategy to control or edit the learned models.
- The strategy seems to be widely applied to various long-aged problems across multiple related societies, e.g., continual learning, explainability, and pruning or unlearning, which are tested in this paper.

**Weaknesses:**

**Weakness 1:** Limited understanding of how the learned knowledge relates to the distributional behaviors of models
- The main weakness of this paper is the limited understanding of how keeping the HiFi part results in keeping the knowledge of learned models. Otherwise, how tuning the HiFi part results in forgetting the specific learned knowledge.
- At the conceptual level of understanding, it is quite convincing that the components showing similar distributional behaviors with the layer outputs are probably the crucial parts of the knowledge. However, it is not guaranteed theoretically.

**Weakness 2:** Insufficient quality of presentation and writing
- I strongly believe this venue requires the highest presentation and writing quality. However, the submitted version contains too many grammar errors, unpolished sentences, and low-clarity visualizations, as follows:
- At line 47: a missing full name of 'CNN'
- At many parts: add a whitespace between text and '('
- At many parts: for citations, the form is inconsistent, e.g., at line 166, "behavior (Jia...; Shah et al., (2024))." is correct.
- At line 178: missing comma after i.e.
- At line 185: missing whitespace before "While"
- Figure 2: The size is too small to recognize the plots, formulations, and texts.
- Equation 3: it is better to keep the length within the text width of the page.
- At line 269: keep the name "HiFi"
- At line 328: It seems "Assumption 5" means A1 and A2 at the right upper part. The labeling of assumptions is not matched.
- At line 469: missing punctuation after "Training Details"
- At line 529: "loss" rather than "Loss"
- Figure 5 (in Appendix): The size is too small to recognize the contents.
- I strongly feel that the level of presentations and writing is not reaching the level of this venue.

**Weakness 3:** Limited comparison with other related works
- Although the authors have provided the 'Related Work' part in the Appendix, it seems insufficient to provide deep insights into this work beyond others.
- For instance, beyond the technically similar model editing methods, in-depth analysis of the prior works investigating the importance of weights or sensitivity measures of weights should be considered. I think that HiFi is another viewpoint to measure the importance of weights so that it has the potential to show further impact on continual learning (also without data of the past tasks) and explainability.

**Questions:**

**Question 1:** unclear notations in equations
- In the "What is Model Editing" part on line 176, to my understanding, 'B' is the number of components (not an individual weight, but a group of weights) in the model. Therefore, the equation, $\|\theta\|-B$, looks wrong because $\|\theta\|$ is commonly used for the number of weights, not components. Would you clarify the equations?

---

> ### Author Response · Authors · 2024-11-25
> **Response to reviewer ZvuL**
>
> We thank the reviewer for their detailed feedback, and for appreciating the distributional approach we used, as well as the potentially broad applicability of our approach.
>
> Based on their comments, we have made changes to the manuscript to enhance clarity.
>
> We now address the individual concerns raised by reviewer ZvuL.
>
> >   **W1 A: The main weakness of this paper is the limited understanding of how keeping the HiFi part results in keeping the knowledge of learned models. Otherwise, how tuning the HiFi part results in forgetting the specific learned knowledge.**
>
> In this work, the 'knowledge' refers to the data distributions learned by the model - thus, by the definition of HiFi components (Section 4.1, lines 245-288), keeping HiFi channels in well-trained models ensures that the learned distributions are maintained. The reconstruction error based formulation proposed in Section 4 is principled and rigorously derived.  Thus, by maintaining the HiFi components, it follows that the distributional information regarding the entire data distribution (for pruning), or the remain classes (for unlearning) is maintained.
>
> > **W1 B: At the conceptual level of understanding, it is quite convincing that the components showing similar distributional behaviors with the layer outputs are probably the crucial parts of the knowledge. However, it is not guaranteed theoretically.**
>
> We point out that the knowledge learned by the model is the distribution generating the data, and as such, retaining HiFi components ensures that the model's knowledge is retained.  Since *retaining* the HiFi components are  **rigorously derived** from the minimization of the layer-wise output reconstruction error, our CoBRA-P and CoBRA-U algorithms (Section 6) are principled, theoretically motivated approach that ensures that knowledge is retained (CoBRA-P) or removed (CoBRA-U) as needed.
>
> > **W2: Quality of presentation**
>
> We thank the reviewer for the feedback. We have addressed the typographical errors in the work, and have fixed the size of images in Figure 5. Additionally, we have made other improvements to the text, highlighted in blue.
>
> > **W3 A: Although the authors have provided the 'Related Work' part in the Appendix ... in-depth analysis of the prior works investigating the importance of weights or sensitivity measures of weights should be considered.**
>
> The setting of our work, wherein we need to *edit* models (specifically for pruning and classwise unlearning), without access to the training data or loss function, is unique, with very few technically similar works available. We point out to the reviewer our significant literature survey into structured pruning methods, most of which analyze the importance of weights. For instance, in Appendix A.3, we discuss works that use gradients to measure weight importance for structured pruning [Molchanov et al, 2019], Weight Norms [Li et al, 2019], reconstruction error [Luo et al, 2017], feature map ranks [Lin et al, 2020], and a variety of other methods (see the survey [Hoefler et al, 2021]). In Appendix A.2, we note [Wang et al, 2022], which uses class-discriminative scores for unlearning in the federated setting.  However, we have clarified Appendix A from Lines 894 to 906 to further highlight different measures of weight importance used in the literature.
>
> > **W3 B: I think that HiFi is another viewpoint to measure the importance of weights so that it has the potential to show further impact on continual learning (also without data of the past tasks) and explainability.**
>
> We are glad that you point out that our notion of HiFi components is applicable other important model editing tasks. However, the scope of this work is limited to using HiFi components to combine structured pruning and classwise unlearning under the model editing umbrella.
>
> > **Q1: In the "What is Model Editing" part on line 176, to my understanding, 'B' is the number of components (not an individual weight, but a group of weights) in the model. Therefore, the equation,  |θ|−B, looks wrong because  |θ|  is commonly used for the number of weights, not components. Would you clarify the equations?**
>
> $|\theta|$ should indeed represent the number of components in the model. We have amended the main document by defining $C_{total}$, the total number of components in the network in place of $|\theta|$.  The change is highlighted in blue.

---

> ### Author Response · Authors · 2024-11-26
> **Summarizing rebuttal and invitation for discussion**
>
> The reviewer has noted the strengths of this work and raised valuable concerns which we have adressed in our rebuttal.
>
> - We have made several corrections to enhance the clarity in presentation
> - We have added additional context in the Related work (Appendix A) to incorporate the feedback of the reviewer
> - Addressed the individual weaknesses in the [General Rebuttal](https://openreview.net/forum?id=wLR9d5ZFpY&noteId=31I6DjaSLz) and the [Respose to the reviewer](https://openreview.net/forum?id=wLR9d5ZFpY&noteId=cQbDp25ozd)
>
> With the announcement of the *extension of the review period*, and several modifications to improve the draft, we hope to engage with the reviewer in a discussion to address their concerns in raising their score for this work.

---

> ### Author Response · Authors · 2024-11-30
> **Invitation for discussion**
>
> Dear Reviewer ZvuL,
>
> We would like to remind you that the discussion period has been extended and there are **3 days** left in the author-reviewer discussion period.
> We are eager to engage with you to address your concerns. We have posted our [General Response](https://openreview.net/forum?id=wLR9d5ZFpY&noteId=31I6DjaSLz), and [individual response](https://openreview.net/forum?id=wLR9d5ZFpY&noteId=cQbDp25ozd) with a [summary](https://openreview.net/forum?id=wLR9d5ZFpY&noteId=Uz7MW43rwo).

---

### Official Review · Reviewer_gwP1 · 2024-11-04

**Soundness:** 3
**Presentation:** 1
**Contribution:** 2
**Rating:** 3
**Confidence:** 3

**Summary:**

This paper addresses the problem of model editing (specifically, structured pruning and class unlearning) for deep neural networks when training data is not inaccessible. The authors propose the concept of "HiFi components", which are identified as a small subset of channels in each layer being responsible for the model's output. Detecting "HiFi components" could be solved by measuring the reconstruction error of these channels. However, due to the unavailable training data, the authors propose a heuristic "RowSum" to identify the similarity between distributions of input contribution and output feature map in a layer. Then HiFi components are the components having a high correlation(/similarity) between input channel contributions and the output feature map. To restore the model's accuracy after editing, the authors derive an algorithm called "BNFix" to update BN's statistics using only distributional access to the data distribution. Two algorithms COBRA-P and COBRA-U are proposed to find whether retaining or discarding HiFi components in pruning and unlearning, respectively. Empirical evaluations on CIFAR-10/100 and ImageNet datasets show the effectiveness of their approach in maintaining competitive accuracy.

**Strengths:**

1. The paper tackles the problem of model editing without accessible training data for the circumstances of structure pruning and class unlearning.

2. Identifying the HiFi component with the proposed correlation measure is interesting to me.

**Weaknesses:**

1. While the concept of HiFi components is interesting, the technical novelty of the RowSum heuristic and BNFix algorithm appears limited. There are many papers proposing to update BN's parameters, a similar strategy to the one in this paper.

2. The theoretical analysis focuses on providing upper bounds on the loss function, however, K is the largest eigenvalue of the hessian, which might not be tight enough as a guarantee.

3. The overall writing and organization of the paper could be improved significantly. The presentation of the main framework and the transition between different concepts in sections should be intuitive.

**Questions:**

1. In Section Introduction, How do photos from a personal device constitute samples of a large collection of photos having similar distributions?

2. "In Figure 2, we show the relative reconstruction error after removing filters from a selection of layers of a ResNet50 trained on CIFAR10". Could you explain how to get Fig. 2 in detail, which is a key assumption in this paper?

3. The introduction of HiFi components and the section of BN fix seem disjointed. Could you provide a clearer connection between these two concepts and explain why only BN's statistics are fixed?

4. Is there anything additional information that needs to be stored during training time for the proposed methods to work?

5. The empirical evidence is primarily based on CIFAR-10/100 and ImageNet datasets, and it would be beneficial to evaluate the methods on more datasets and tasks.

6. There are many citation errors in the text. Please carefully check. The font of the figures is really tiny,  making them very difficult to read.

---

> ### Author Response · Authors · 2024-11-25
> **Response to reviewer gwP1**
>
> We thank the reviewer for their time and noting the difficulty of the problem setting and our approach to identifying HiFi components novel.
>
> Based on their comments, we have made changes to the manuscript to enhance clarity with changes highlighted in blue.
>
> We now address the individual concerns raised by reviewer gwP1.
>
> > **W1 A: While the ... technical novelty of the RowSum heuristic and BNFix algorithm appears limited.**
>
> **RowSum:** This work is the first to propose the notion of HiFi components in lines 228-230 to identify components that are responsible for making predicitions in a challenging setup- in models with complex interconnections without access to training data or a loss function. The RowSum heuristic is proposed as a feasible technique in this setting to identify HiFi channels. As also noted by reviewer ZvuL, this technique is broadly applicable beyond the problems of pruning and editing.
>
> **BNFix:** As noted as a strength of this work by reviewer d7eY, this work is the first to rigorously analyze BatchNorm at inference utilizing advanced tools such as Fact 1 and state the update as an algorithm in Algorithm 2. As demonstrated by our experiments in pruning and unlearning presented in Tables 1, and 2, BNFix is an effective replacement for retraining in our setting. Additionally, our work is also the first to highlight the impact of rectifying BatchNorm statistics on Unlearning, as shown in Table 2. Additionally, we derive other algorithms (Algorithm 4) based on our anaysis in Appendix B.9.
>
> > **W1 B: There are many papers proposing to update BN's parameters, a similar strategy to the one in this paper.**
>
> We wish to clarify that we do not update BatchNorm _learned parameters_ $\gamma$ and $\beta$, only adjusting the *means and variances* $\mu$ and $\sigma$ as per Algorithm 2. We would like to emphasize the setting in which this update is proposed-editing without access to training data or loss functions. We kindly request the reviewer to mention any other papers that propose a similar strategy in this setting that we may have missed.
>
> > **W2: The theoretical analysis focuses on ... might not be tight enough as a guarantee.**
>
> The focus of our theoretical analysis in Lemma 1 and Theorem 1 is not to obtain bounds that are strictly tight but to obtain viable procedures which can emperically validated.
> We would like to highlight that our bound is informative in explaining the improvement in the model’s accuracy after rectifying the BatchNorm statistics as shown in Appendix B.2 and B.5.
> In Appendix B.9, we discuss Algorithm 4, a variant based on our theoretical analysis in Theorem 1 and emperically verify the effectiveness of this variants, indicating that our analysis is informative.
> Moreover, the theoretical analysis using the largest eigenvalue of the Hessian follows standard practice, which assumes that the loss function is Lipschitz smooth, including other works analyzing BatchNorm, such as [1].
>
> We address the questions in the subsequent comment.
>
> > References
> 1. How does BatchNorm help Optimization Santurkar et al., 2018

---

> ### Author Response · Authors · 2024-11-25
> **Response to reviewer gwP1, continued**
>
> > **Q1: In Section Introduction, How do photos from a personal device constitute samples of a large collection of photos having similar distributions?**
>
>  Our method is inspired by works such as [2], which motivates the use of samples drawn from a similar distribution to overcome concerns of privacy and security. Photos from a personal device capture the same underlying distribution as a training set, which is a large collection of curated images. For example, a large repository of cat images and the photos of cats from a cat owner’s device both capture features relating to cats. This is one example of distributional access, in Section 7, we use 1500 synthetic samples from generative models as described in Appendix B.1.
>
> > **Q2: In Figure 2, we show the relative reconstruction error after removing filters from a selection of layers of a ResNet50 trained on CIFAR10". Could you explain how to get Fig. 2 in detail, which is a key assumption in this paper?**
>
> In Figure 2, we compute the distributional dissimilarity (Equation 3) between each input contribution to a layer (post-activation features generated by filters in the previous layer), and the aggregate feature map (which is the sum of the input contributions). We have amended the manuscript to reflect this difference in Lines 260-268, and have clarified the plots to highlight which components are considered HiFi. We observe that only between 5 and 30% of filters are HiFi, which motivates our model editing strategy: identify HiFi components and keep them (pruning) or remove them (unlearning).
>
> > **Q3 A: The introduction of HiFi components and the section of BNfix seem disjointed. Could you provide a clearer connection between these two concepts**
>
> The recipie for effective model editing described in Algorithm 1 requires editing and recovery which are independent aspects. Identifying HiFi channels addresses the challenge of finding components responsible for prediction in the setting of this work - editing models with complex interconnections without the training data or loss function.
> This setting also necessitates alternatives for retraining models that do not require the loss function or training data. BNFix is identified as a suitable replacement for retraining in this setting.
>
> > **Q3 B: explain why only BN's statistics are fixed?**
>
> The BNFix algorithm can also be derived by analyzing the expected reconstruction error after a BatchNorm layer; we have added this derivation in Appendix E. In the main work, we present Theorem 1 as it quantifies the effect of adjusting the BatchNorm statistics on the loss function, as opposed to analyzing the local behaviour.
>
> >**Q4:  Is there anything additional information that needs to be stored during training time for the proposed methods to work?**
>
> No. The strength of our proposed technique is that our procedures are agnostic to the training procedure of the models. Our techniques can be applied to pretrained models and do not require storing any additional information at training time.
>
> >**Q5: The empirical evidence is primarily ... evaluate the methods on more datasets and tasks.**
>
> Our technique is the first to tackle the setting of model editing for pruning and class unlearning. Our empirical evaluation follows the settings proposed in the state of the art techniques for pruning and unlearning [3,4], which focus on standard image classification datasets. More complicated tasks are the subject of our future investigations.
>
> > **Q6: There are many citation errors ...**
>
> We thank the reviewer for noting these corrections, we have incorporated these changes into manuscript.
>
> > References
> 1. How does BatchNorm help Optimization Santurkar et al., 2018
> 2.  Dreaming to Distil, Yin et al., 2021.
> 3. DFPC: Data flow driven pruning of coupled channels without data, Narshana et al., 2023
> 4. Model sparsity can simplify unlearning, Jia et al., 2023

---

> ### Author Response · Authors · 2024-11-26
> **Rebuttal summary and invitation for discussion**
>
> The reviewer has noted some of the strengths of this work and we have adressed their concerns in our rebuttal.
>
> - We have made several corrections to enhance the clarity in presentation in the updated manuscript
> - We have emphasized on the strengths missed by the reviewer and addressed weaknesses in the [General Rebuttal](https://openreview.net/forum?id=wLR9d5ZFpY&noteId=31I6DjaSLz) and the [Respose to the reviewer](https://openreview.net/forum?id=wLR9d5ZFpY&noteId=1xtivbL2ao)
> - We have answered all of the questions raised by the reviewer in the [Response to the reviewer](https://openreview.net/forum?id=wLR9d5ZFpY&noteId=VbjsE80i4Z)
>
> With the *extension of the review period*, and several modifications to improve the draft, we invite the reviewer to engage in a discussion to address their concerns in raising their score for this work.

---

> ### Author Response · Authors · 2024-11-30
> **Invitation for discussion**
>
> Dear Reviewer gwP1,
>
> We would like to remind you that the discussion period has been extended and there are **3 days** left in the author-reviewer discussion period.
> We are eager to engage with you to address your concerns. We have posted our [General Response](https://openreview.net/forum?id=wLR9d5ZFpY&noteId=31I6DjaSLz), and individual responses([1](https://openreview.net/forum?id=wLR9d5ZFpY&noteId=1xtivbL2ao), [2](https://openreview.net/forum?id=wLR9d5ZFpY&noteId=VbjsE80i4Z)) with a [summary](https://openreview.net/forum?id=wLR9d5ZFpY&noteId=SCi6VaQ2cA).

---

### Author Response · Authors · 2024-11-25
**General Response to reviewers**

We thank the reviewers for their detailed feedback, and offer the following clarifications.


## Clarifications and Overlooked aspects of our work


We take this opportunity to provide clarity on some points that reviewer's may have overlooked, potentially leading to negative appraisals of our work. In particular:

- We wish to highlight that the CoBRA framework is among the first to address **both structured pruning and classwise unlearning** without requiring either the training data or the loss function, and without requiring retraining. We use the local similarities between input contributions of feature maps as a proxy for the loss function.
- In our framework, both pruning and unlearning can be addressed using only distributional information in the form of feature map statistics, specifically the correlations between feature maps.
- In Section 7, we highlight the remarkable recovery of accuracy just by adjusting BatchNorm statistics. Notably, for classwise unlearning, the **forget class accuracy increases by over 90%** without fine-tuning, simply by adjusting BatchNorm statistics **using only remain class samples**.  Moreover, the BNFix algorithm is formally derived (section 5), with a rigorous guarantee on the change in the loss function's value.
- To further address the inability to fine-tune models after editing, we derive a **weight compensation scheme with a formal guarantee on expected reconstruction error** in Theorem 2. Our method ensures that the **distribution of the edited features** is similar to the unedited ones, upto a factor dependent on the covariance of the feature distribution.

To offer further clarification, we summarize the critical aspects of our work below.

**Our work focuses on model editing in the under-researched setting where the training data and loss function are not available, which makes identifying components that are necessary for predictions challenging. We clarify this in detail in the sequel.**

- **Problem Setting**: This work addresses **model editing** - specifically pruning and unlearning - for models with complex interconnections, in the challenging and under-researched setting where the **training data and loss function are not available.** Addressing these problems requires identifying **which components in a network contribute to the model's predictions**.
- **Challenge - Editing Models without Training Data or Loss Functions:**  The lack of the loss function and the training data creates three challenges.
   - **Identifying Important Components without the loss function:** This necessitates identifying components based on **local information**,  as without access to the loss function, global phenomena cannot be measured. In this work, we identify important components as those that generate features that are distributionally similar to aggregate feature maps, called **HiFi components**. We identify them using  **correlations between output features**, specifically with Equation 3.
   - **No training data:** This requires us to leverage **distributional information** instead of training data. In this work, the **distributional approach** we use access to **samples drawn from the same/similar distribution**. Specifically, we use synthetic data as this proxy.
   - **No Fine-Tuning:** As we have no access the loss function, we cannot fine-tune the model after editing, motivating our Fidelity Compensation (Theorem 2) and BNFix (Algorithm 2).

- **Analysis of BatchNorm:** Our theoretical analysis (Theorem 1), which helps explain both these phenomena, is the **first that connects the BatchNorm statistics to the generalization ability of the network**. This provides a principled way to derive the BNFix algorithm, and offers an explanation for the remarkable accuracy recovery after adjusting BatchNorm statistics.


## Weaknesses pointed out by reviewers

To address the concerns raised by the reviewers, we provide the following response.

- All three reviewers expressed concerns with the presentation of our work. To that end, we have incorporated the feedback from all three reviewers into the amended manuscript.

**We address specific concerns in the individual responses to the reviewers.**

---

### Meta-Review · Area_Chair_zcxQ · 2024-12-22

**Metareview:**

This paper tackles the challenge of model editing, specifically structured pruning and class unlearning, for deep neural networks when the training data is inaccessible and the loss function is unknown. The authors introduce the concept of "HiFi components," a small subset of channels within each layer identified as being crucial to the model's output. Their approach involves freezing the HiFi components and adjusting batch normalization to compensate for changes in distributional behavior. To validate their method, the authors present two use cases: pruning and unlearning.

The paper's strengths lie in the novel setup and the introduction of HiFi components, which offer an intriguing perspective. However, the work has notable weaknesses, including limited empirical validation. Specifically, (1) the effectiveness of the learned importance weights compared to prior methods is not sufficiently demonstrated or interpreted, and (2) the proposed method's performance is not adequately benchmarked against baselines in contemporary Transformer-based architectures.

Given these limitations, I recommend rejecting this submission.

**Additional Comments On Reviewer Discussion:**

During the discussion period, Reviewers gwP1 and ZvuL did not respond, while Reviewer d7ey raised their score twice.

I carefully reviewed all concerns raised by the reviewers and found that the authors did not adequately address several critical issues, including some highlighted by Reviewer d7ey.

Specifically, Reviewer ZvuL questioned how the proposed weight importance differs from prior measures of weight importance or sensitivity in terms of their influence on downstream task performance, and potential interpretation. Unfortunately, the authors did not address this point at all.

For Reviewer d7ey's concern about the applicability of the method to Transformer-based architectures, while the authors provided performance evaluations, they failed to compare their results against any baselines. Notably, Reviewer d7ey did not advocate for the acceptance of this work.

These unresolved concerns significantly limit the scope and potential audience for this work. As such, I believe it does not meet the high standards expected for the prestigious ICLR conference.

---

### Decision · Program_Chairs · 2025-01-22

Reject